# Provably Efficient Neural GTD Algorithm for Off-policy Learning

**Hoi-To Wai**
The Chinese University of Hong Kong
Shatin, Hong Kong
htwai@se.cuhk.edu.hk

**Zhuoran Yang**
Princeton University
Princeton, NJ, USA
zy6@princeton.edu

**Zhaoran Wang**
Northwestern University
Evanston, IL, USA
zhaoranwang@gmail.com

**Mingyi Hong**
University of Minnesota
Minneapolis, MN, USA
mhong@umn.edu

## Abstract

This paper studies a gradient temporal difference (GTD) algorithm using neural network (NN) function approximators to minimize the mean squared Bellman error (MSBE). For off-policy learning, we show that the minimum MSBE problem can be recast into a min-max optimization involving a pair of over-parameterized primal-dual NNs. The resultant formulation can then be tackled using a neural GTD algorithm. We analyze the convergence of the proposed algorithm with a 2-layer ReLU NN architecture using $m$ neurons and prove that it computes an approximate optimal solution to the minimum MSBE problem as $m \to \infty$.

## 1 Introduction

Policy evaluation is a key problem in reinforcement learning (RL) whose goal is to estimate the value function of a given policy, i.e., the expected total reward of a discounted Markov decision process (MDP) starting from a given state. Among others, the temporal difference (TD) learning algorithm [Sutton, 1988] has been used to minimize the mean squared (projected) Bellman error (MSBE). For off-policy learning where the behavior policy differs from the target policy, gradient-based TD (GTD) learning algorithms [Sutton et al., 2009a,b] have been proposed with guaranteed convergence. A growing trend is to employ nonlinear approximation such as neural network (NN) functions [e.g., Chung et al., 2019, Haarnoja et al., 2018, Lillicrap et al., 2015, Mnih et al., 2016, Silver, 2012].

The TD/GTD learning algorithms have been analyzed with *linear function approximation* [Bhandari et al., 2018, Dalal et al., 2017]. Meanwhile, nonlinear TD/GTD learning algorithms (as well as other related problems such as Q-learning) are studied in [Bhatnagar et al., 2009, Brandfonbrener and Bruna, 2019, Chung et al., 2019, Dai et al., 2018, Wai et al., 2019], also see [Bertsekas, 2019]. However, these algorithms lack theoretical guarantees related to minimizing the MSBE or MSPBE objective as they may get stuck in a local optimum. Furthermore, these algorithms are designed for arbitrary nonlinear function approximation, whose actual implementations involve computationally intensive steps such as computing the Hessians for the nonlinear functions.

In this paper, we analyze the efficiency of an off-policy GTD learning algorithm with NN function approximation. We consider a simplified setting employing a pair of over-parameterized 2-layer ReLU NNs. A key result proven is that the proposed neural GTD algorithm is guaranteed to *converge globally* to a minimizer of the MSBE problem, despite the corresponding optimization problem is non-convex. Our main contributions are:

- We derive a new formulation for MSBE minimization with a *primal NN* and a *dual NN* as approximators. A neural GTD algorithm, which obeys similar update rules as the classical GTD2 algorithm, is proposed for the resultant min-max optimization problem.
- Under an off-policy learning setting, we analyze the convergence rates of the neural GTD algorithm with various sampling techniques, including population update, stochastic updates with i.i.d. samples and Markov samples.
- We focus on the 2-layer ReLU NN architecture. We show that when the width of the NN employed goes to infinity and the TD error function lies in the NN function class, the proposed neural GTD algorithm is guaranteed to find a *global minimizer* of the MSBE problem. Importantly, the convergence rates measured with the functional distance are independent of NN's width.

**Related Works**   Recent works have studied the *global convergence to optimal solutions* of different RL algorithms. Examples are [Cai et al., 2019, Wang et al., 2019, Xu and Gu, 2019] which studied neural TD learning, neural policy gradient, and neural Q-learning, respectively. These works rely on that in the limit when the number of neurons approaches infinity, the *nonlinear* NN function is locally approximated by a *linear* function. The present paper follows a similar philosophy, i.e., by reducing the neural GTD algorithm to a primal-dual method for solving a convex-concave problem. We develop new analysis to handle biases in gradients and off-policy learning settings.

Our work is also related to recent works on finite time guarantees for GTD learning algorithms using linear function approximation. For instance, Dalal et al. [2018, 2019], Liu et al. [2015] provide guarantees when the algorithms acquire i.i.d. samples; Du et al. [2017] apply variance reduction technique; Doan [2019], Gupta et al. [2019], Wang et al. [2017], Xu et al. [2019] consider when Markov samples are used. In comparison, we extend these analysis to using NN function approximation and offer finite-time guarantees in the overparameterization limit.

Lastly, the present work is related to recent advances in overparameterized (deep) NNs. Inspired by the classical works [Bartlett, 1998, Rahimi and Recht, 2008] and empirical studies in [Neyshabur and Li, 2019, Zhang et al., 2016], it has been recently shown that overparameterized NNs are efficiently learnable using gradient-type algorithms in [Allen-Zhu et al., 2019a,b, Arora et al., 2019, Jacot et al., 2018, Lee et al., 2019, Mei et al., 2018]. A key insight used is that NN functions exhibit an implicit local linearization, which allows one to 'convexify' the corresponding training problem and establish global convergence. This line of analysis is also known as 'lazy training' for NNs. Notice that the recent works [Allen-Zhu and Li, 2020, Chizat and Bach, 2018, Daniely, 2017] have suggested stronger theories for the generalization power of deep NNs beyond the 'lazy training' characterization. This paper develops the analysis by adapting the above results to neural GTD learning via the implicit local linearization. Unlike the above works, our challenge involves providing dimension-independent bounds in the function space for the off-policy learning algorithm.

## 2   Markov Decision Process

Consider a discounted markov decision process (MDP) described by $(S, A, P^a, R, \gamma)$ where $S$ is the possibly infinite state space and $A$ is the action space. Given an action $a \in A$, the operator $P^a : S \times S \to \mathbb{R}_+$ is a Markov transition kernel such that for any measurable function $f$ on $S$, we denote for any $s \in S$ that

$$\mathbb{E}\big[f(s')|s' \sim P^a(s, \cdot)\big] = P^a f(s) = \int_S f(y) P^a(s, \mathrm{d}y), \tag{1}$$

where $s'$ denotes the next state that is transitioned into. The function $R(s, a)$ is the reward received after action $a$ in state $s$; lastly, $\gamma \in (0, 1)$ is the discount factor. We assume $\sup_{s,a} R(s, a) \le \overline{r}$.

The *target policy* $\pi(a|s)$ is the conditional probability of action $a \in A$ given the current state $s \in S$ [Szepesvári, 2010]. This induces a Markov chain with the transition kernel $P^\pi(s, \cdot) := \mathbb{E}_{a \sim \pi(\cdot|s)}[P^a(s, \cdot)]$. The *policy evaluation* problem aims at learning a *value function* $V^\pi : S \to \mathbb{R}$, defined as the discounted total reward starting from state $s$:

$$V^\pi(s) := \mathbb{E}\big[\textstyle\sum_{t=0}^\infty \gamma^t R(s_t, a_t) \,\big|\, s_0 = s, \ a_t \sim \pi(\cdot|s_t), \ s_{t+1} \sim P^{a_t}(s_t, \cdot)\big]. \tag{2}$$

Applying the Bellman equation [Van Hasselt, 2012] shows that

$$0 = V^\pi(s) - \mathbb{E}_{a \sim \pi(\cdot|s)}\big[R(s, a)\big] - \gamma P^\pi V^\pi(s). \tag{3}$$

The existence of $V^\pi(s)$ follows if the Markov chain driven by $P^\pi$ is aperiodic and irreducible. From the above, the policy evaluation problem may be solved by finding a value function $V^\pi(s)$ which

satisfies (3). This problem is challenging since (i) the state space $\mathsf{S}$ is large, and (ii) the transition kernel $\mathrm{P}^\pi(\cdot, \cdot)$ is unknown. The latter must be learnt from the observed state/action pairs while solving the policy evaluation simultaneously.

## 2.1 Off-policy Policy Evaluation with Function Approximation

We consider *nonlinear* approximation of the value function using two-layer neural network (NN) of $m$ neurons with rectified linear units (ReLUs). Each state $s \in \mathsf{S}$ is encoded by a $d$-dimensional vector, denoted by $\boldsymbol{x}_s \in \mathbb{R}^d$ with $\|\boldsymbol{x}_s\|_2 = 1$. The NN is over-parameterized with $m \gg 1$ neurons, each with the weight as $b_r \in \mathbb{R}$, and the parameter described by $\boldsymbol{\theta} \in \Theta \subseteq \mathbb{R}^{md}$ (for simplicity, we may assume $\Theta = \mathbb{R}^{md}$). Our aim is to approximate the value function in (3) as:

$$\mathrm{V}^\pi(s) \approx f(s, \boldsymbol{\theta}) := \sum_{r=1}^m \frac{b_r}{\sqrt{m}} \mathrm{ReLU}(\langle \boldsymbol{\theta}^{(r)}, \boldsymbol{x}_s \rangle) = \sum_{r=1}^m \frac{b_r}{\sqrt{m}} \mathbb{1}\{\langle \boldsymbol{\theta}^{(r)}, \boldsymbol{x}_s \rangle > 0\} \langle \boldsymbol{\theta}^{(r)}, \boldsymbol{x}_s \rangle, \quad (4)$$

where $\mathbb{1}\{\cdot\}$ is the 0-1 indicator function with $\mathbb{1}\{\mathcal{E}\} = 1$ if $\mathcal{E}$ is true; otherwise $\mathbb{1}\{\mathcal{E}\} = 0$. Notice that $f(\cdot, \boldsymbol{\theta})$ is differentiable with respect to $\boldsymbol{\theta}$ when $\langle \boldsymbol{\theta}^{(r)}, \boldsymbol{x}_s \rangle \neq 0$. We assume:

**H1.** *Consider the NN function* (4). *The weights $b_r$ are generated as $b_r \sim \mathcal{U}(\{-1, 1\})$. The parameters $\boldsymbol{\theta} = (\boldsymbol{\theta}^{(1)}, ..., \boldsymbol{\theta}^{(m)})$ are initialized with $\boldsymbol{\theta}_0^{(r)} \sim \mathcal{N}(\mathbf{0}, \frac{1}{d}\mathrm{I}_d)$. All weights/parameters are drawn independently. These initialization parameters are denoted by $\Xi_0 = (\boldsymbol{\theta}_0, b_1, ..., b_m)$.*

Fixing the NN weights at $(b_r)_{r=1}^m$. Let $B > 0$ be a fixed radius, we consider the NN function approximation (4) where $\boldsymbol{\theta}$ is taken from the ball $S_B = \{\boldsymbol{\theta} \in \Theta : \|\boldsymbol{\theta} - \boldsymbol{\theta}_0\| \leq B\}$.

**Off-policy Learning**. We consider *off-policy learning*, where the observed state-action triplets $(s, a, s')$ are generated from a *behavior policy* $\Pi(a|s) \neq \pi(a|s)$. Similarly, $\mathrm{P}^\Pi$ is the induced Markov kernel for the state sequence $(s_0, s_1, ...)$. The Markov chain induced by $\mathrm{P}^\Pi$ is assumed ergodic with the unique stationary distribution $\mu^\Pi$, and the supports of policies satisfy $\mathrm{supp}\{\pi(\cdot|\cdot)\} \subseteq \mathrm{supp}\{\Pi(\cdot|\cdot)\}$. Let the importance ratio and the temporal difference (TD) error be

$$\rho(a|s) := \frac{\pi(a|s)}{\Pi(a|s)}, \quad \delta(s, a, s'; \boldsymbol{\theta}) := f(s, \boldsymbol{\theta}) - \gamma f(s', \boldsymbol{\theta}) - \mathrm{R}(s, a). \quad (5)$$

The following conditional expectation approximates the r.h.s. of (3):

$$\bar{\delta}(s, \boldsymbol{\theta}) := \mathbb{E}_{a \sim \Pi(\cdot|s), s' \sim \mathrm{P}^a(s, \cdot)}\big[\rho(a|s)\delta(s, a, s'; \boldsymbol{\theta})\big] = \mathbb{E}_{a \sim \pi(\cdot|s), s' \sim \mathrm{P}^a(s, \cdot)}\big[\delta(s, a, s'; \boldsymbol{\theta})\big]. \quad (6)$$

Our goal is to find the NN parameter $\boldsymbol{\theta} \in S_B$ to minimize the mean squared Bellman error (MSBE). Let $\upsilon \geq 0$ be the regularization parameter, we formulate the MSBE minimization problem

$$\min_{\boldsymbol{\theta} \in S_B} J(\boldsymbol{\theta}) := \mathbb{E}_{s \sim \mu^\Pi}\big[\tfrac{1}{2}\bar{\delta}(s, \boldsymbol{\theta})^2 + \tfrac{\upsilon}{2} f(s, \boldsymbol{\theta})^2\big], \quad (7)$$

When $\upsilon = 0$, one may apply an off-policy modification to the neural TD learning in [Cai et al., 2019], with $\boldsymbol{\theta}' \leftarrow \boldsymbol{\theta} - \beta\rho(a|s)\delta(s, a, s'; \boldsymbol{\theta})\nabla f(s, \boldsymbol{\theta})$, where $\beta > 0$. However, as observed by [Baird, 1995, Sutton et al., 2016], with off-policy samples the TD algorithm may diverge; also see §3.3. This is due to the non-Hurwitz matrix which forms the mean field of update, since the behavior and target policies are mismatched. Below, we develop an algorithm similar to *gradient-based* TD (GTD) learning which is shown to resolve the non-convergent issue.

**Neural GTD Algorithm.** Consider rewriting the MSBE function as:

$$\mathbb{E}_{s \sim \mu^\Pi}\big[\tfrac{1}{2}\bar{\delta}(s, \boldsymbol{\theta})^2\big] = \int_\mathsf{S} \tfrac{1}{2}\bar{\delta}(s, \boldsymbol{\theta})^2 \mu^\Pi(ds) \stackrel{(a)}{=} \int_\mathsf{S} \max_{y(s)} \big\{y(s)\bar{\delta}(s, \boldsymbol{\theta}) - \tfrac{1}{2}y(s)^2\big\} \mu^\Pi(ds)$$
$$\stackrel{(b)}{=} \max_{y(\cdot)} \big\{\mathbb{E}_{s \sim \mu^\Pi}\big[y(s)\bar{\delta}(s, \boldsymbol{\theta})\big] - \tfrac{1}{2}\mathbb{E}_{s \sim \mu^\Pi}[y(s)^2]\big\}, \quad (8)$$

where (a) is due to $\frac{1}{2}\delta^2 = \max_y\{y\delta - \frac{y^2}{2}\}$, and (b) swapped the $\max$ and $\int$ operators. Substituting (8) into (7) yields a non-convex min-max problem. Notice that the above reformulation is inspired by those derived in prior works such as [Dai et al., 2017, 2018, Shapiro, 2011].

The optimizer to the maximization (8) is given by $y^\star(s, \boldsymbol{\theta}) = \bar{\delta}(s, \boldsymbol{\theta})$, which is the expected TD error. Notice that this leads to another intractable problem since $|\mathsf{S}|$ is large. As such, we consider applying an *additional NN function approximation*. We find an NN parameter $\boldsymbol{w} \in S_B$ such that

$$\bar{\delta}(s, \boldsymbol{\theta}) \approx f(s, \boldsymbol{w}), \ \forall \ s \in \mathsf{S}. \quad (9)$$

---

**Algorithm 1** Neural GTD algorithms for MSBE

---

1: **Input**: step sizes $(\beta_k)_{k \geq 0}$; maximum number of iterations $n$.

2: Generate initialization parameters $\boldsymbol{\theta}_0 = \boldsymbol{w}_0 \in \mathbb{R}^{md}$ with H1, and draw a random integer $I_n$:

$$\mathbb{P}(I_n = k) = \beta_k / \sum_{\ell=0}^{n} \beta_\ell, \ k = 0, ..., n. \tag{11}$$

3: **for** $k = 0, 1, 2, ..., I_n$ **do**

4:     *(I.i.d. sample)* Draw a state $s \sim \mu^\Pi$ and set $s_k = s$; or *(Markov sample)* set $s_k = s'_{k-1}$.

5:     Draw action $a_k \sim \Pi(\cdot|s_k)$, and $s'_k \sim \mathrm{P}^{a_k}(s_k, \cdot)$ according to the behavior policy.

6:     Compute the gradient at $\boldsymbol{\theta}_k$ as $\nabla\widetilde{\delta}_k(\boldsymbol{\theta}_k)$, where $\widetilde{\delta}_k(\boldsymbol{\theta}_k)$ is the empirical TD error:

$$\widetilde{\delta}_k(\boldsymbol{\theta}) := \rho(a_k|s_k)\left\{ f(s_k, \boldsymbol{\theta}) - \gamma f(s'_k, \boldsymbol{\theta}) - \mathrm{R}(s_k, a_k) \right\}. \tag{12}$$

7:     Let $\mathcal{P}_{S_B}(\cdot)$ be the Euclidean projection onto $S_B$, perform the updates as:

$$\begin{aligned}
\boldsymbol{\theta}_{k+1} &= \mathcal{P}_{S_B}\left\{ \boldsymbol{\theta}_k - \beta_k f(s_k, \boldsymbol{w}_k)\nabla\widetilde{\delta}_k(\boldsymbol{\theta}_k) - \beta_k v f(s_k, \boldsymbol{\theta}_k)\nabla f(s_k, \boldsymbol{\theta}_k)) \right\}, \\
\boldsymbol{w}_{k+1} &= \mathcal{P}_{S_B}\left\{ \boldsymbol{w}_k + \beta_k \widetilde{\delta}_k(\boldsymbol{\theta}_k)\nabla f(s_k, \boldsymbol{w}_k) - \beta_k f(s_k, \boldsymbol{w}_k)\nabla f(s_k, \boldsymbol{w}_k)) \right\}.
\end{aligned} \tag{13}$$

8: **Return**: (approx.) optimal parameters $(\boldsymbol{\theta}_{I_n}, \boldsymbol{w}_{I_n})$.

---

Using (9), problem (7) is approximated as a *non-convex non-concave* min-max problem:

$$\min_{\boldsymbol{\theta} \in S_B} \max_{\boldsymbol{w} \in S_B} \ J_v(\boldsymbol{\theta}, \boldsymbol{w}) := \mathbb{E}_{s \sim \mu^\Pi}\left[ f(s, \boldsymbol{w})\bar{\delta}(s, \boldsymbol{\theta}) - \frac{1}{2}f(s, \boldsymbol{w})^2 + \frac{v}{2}f(s, \boldsymbol{\theta})^2 \right]. \tag{10}$$

Observe that problem (10) involves the simultaneous optimization of *two NNs*. In addition to the 'primal NN' ($\boldsymbol{\theta}$), we employ a 'dual NN' ($\boldsymbol{w}$) to approximate the TD error (5). We propose a neural GTD algorithm in Algorithm 1. The algorithm, which shares similar update equations as GTD2 [Sutton et al., 2009a], is essentially a projected primal-dual gradient method on (10). Compared to nonlinear algorithms such as [Bhatnagar et al., 2009], the neural GTD algorithm does not involve computing the Hessians of NN which makes it more practical for the over-parameterized setting.

**Remark 1.** *Dai et al. [2018] derived a similar reformulation of the MSBE function with nonlinear function approximation as (10). However, unlike the neural GTD which is a single loop algorithm, [Dai et al., 2018, Algorithm 1] requires solving an inner maximization for each iteration which possibly requires a simulator. Furthermore, the analysis therein is not suitable for the overparameterized NN setting as it involves bounding the approximation error using an $\ell_\infty$ norm.*

**Remark 2.** *We notice that the GTD2 algorithm [Sutton et al., 2009a] shares a similar update equation as the neural GTD, yet GTD2 was derived through minimizing a projected MSBE objective function. This coincidence can be justified as the approximation in (9) is exact when the expected TD error lies in the space of NN functions. As such, the latter implicitly imposed a function projection. A key difference between our derivation and an algorithm derived from an exact projected MSBE formulation is that the ours avoids a Hessians computation step; e.g., see [Bhatnagar et al., 2009].*

## 3   Finding a Global Minimizer for MSBE

This section summarizes our main results. Our analysis hinges on the following linearized NN function:

$$\widehat{f}(s, \boldsymbol{\theta}) := \sum_{r=1}^{m} \frac{b_r}{\sqrt{m}} \mathbb{1}\left\{ \langle \boldsymbol{\theta}_0^{(r)}, \boldsymbol{x}_s \rangle > 0 \right\} \langle \boldsymbol{\theta}^{(r)}, \boldsymbol{x}_s \rangle \equiv \langle \boldsymbol{\theta}, \ell(\boldsymbol{x}_s) \rangle, \tag{14}$$

where $\ell : \mathbb{R}^d \to \mathbb{R}^{md}$. Compared to (4), the function $\widehat{f}(s, \boldsymbol{\theta})$ is identical to $f(s, \boldsymbol{\theta})$ except for fixing $\boldsymbol{\theta} = \boldsymbol{\theta}_0$ in the activation function $\mathbb{1}\{\cdot\}$. Moreover, $\widehat{f}(s, \boldsymbol{\theta})$ is differentiable w.r.t. $\boldsymbol{\theta}$. Note that $\|\ell(\boldsymbol{x}_s)\|_2^2 \leq 1$ for any $s \in \mathsf{S}$ and initialization $\Xi_0$ [cf. H1].

Our central idea is to treat $\widehat{f}(s, \boldsymbol{\theta})$ as a surrogate to the *nonlinear* NN function $f(s, \boldsymbol{\theta})$. In particular, we analyze the convergence of neural GTD for finding a saddle point of the linearized problem:

$$\min_{\boldsymbol{\theta} \in S_B} \max_{\boldsymbol{w} \in S_B} \widehat{J}_v(\boldsymbol{\theta}, \boldsymbol{w}) := \mathbb{E}_{s \sim \mu^\Pi}\left[ \widehat{f}(s, \boldsymbol{w})\widehat{\delta}(s, \boldsymbol{\theta}) - \frac{1}{2}\widehat{f}(s, \boldsymbol{w})^2 + \frac{v}{2}\widehat{f}(s, \boldsymbol{\theta})^2 \right], \tag{15}$$

where

$$\widehat{\delta}(s,\boldsymbol{\theta}) := \mathbb{E}_{a\sim\Pi(\cdot|s),s'\sim\mathrm{P}^a(s,\cdot)}\big[\rho(a|s)\big\{\widehat{f}(s,\boldsymbol{\theta}) - \gamma\widehat{f}(s',\boldsymbol{\theta}) - \mathrm{R}(s,a)\big\}\big], \tag{16}$$

Problem (15) differs from (10) through linearizing the NN functions. Importantly, problem (15) is a convex-concave optimization with the saddle point denoted as $\widehat{\boldsymbol{z}}(\Xi_0) = (\widehat{\boldsymbol{\theta}}(\Xi_0), \widehat{\boldsymbol{w}}(\Xi_0))$.

Assuming that $m \gg 1$, in Section 3.1, we show that the neural GTD algorithm converges to a saddle point of (15); then in Section 3.2, we show that an optimal solution to (7) can be taken as the primal solution in a saddle point to (15). Finally, Corollary 3.1 shows that the neural GTD algorithm finds a global minimizer of MSBE.

### 3.1 Convergence to Saddle Point of (15)

We present convergence guarantees for neural GTD in two favors — first we study a population-based method using exact gradients; then we study the stochastic methods using i.i.d. and Markov samples. Before proceeding, we state the following assumption on the stationary distribution:

**H2.** *There exists a constant $c_0$ such that for any $\tau > 0$, $\boldsymbol{y} \sim \mathcal{N}(0, \frac{1}{d}\mathrm{I}_d)$, it holds almost surely that*

$$\mathbb{E}_{s\sim\mu^\Pi}\big[\mathbb{1}\{|\langle\boldsymbol{y},\boldsymbol{x}_s\rangle| \le \tau\}|\boldsymbol{y}\big] \le c_0\tau/\|\boldsymbol{y}\|_2. \tag{17}$$

The above condition is a regulatory assumption which requires $\boldsymbol{x}_s$ to be 'uniformly' distributed when $s$ is drawn from $\mu^\Pi$. To simplify notations, we let $\boldsymbol{z}_k := (\boldsymbol{\theta}_k^\top\ \boldsymbol{w}_k^\top)^\top$. Also, we define the $L^2$ distance

$$d_{\widehat{f}}(\boldsymbol{z}_k, \widehat{\boldsymbol{z}}(\Xi_0)) := \mathbb{E}_{s\sim\mu^\Pi}\Big[|\widehat{f}(s,\boldsymbol{\theta}_k) - \widehat{f}(s,\widehat{\boldsymbol{\theta}}(\Xi_0))|^2 + |\widehat{f}(s,\boldsymbol{w}_k) - \widehat{f}(s,\widehat{\boldsymbol{w}}(\Xi_0))|^2\Big], \tag{18}$$

where $\widehat{\boldsymbol{z}}(\Xi_0)$ is a saddle point to (15) given the initial NN parameters $\Xi_0$. If $d_{\widehat{f}}(\boldsymbol{z}_k, \widehat{\boldsymbol{z}}(\Xi_0)) \approx 0$, then $\boldsymbol{z}_k$ gives the primal-dual NNs that are close in the function space to the NNs parameterized by $\widehat{\boldsymbol{z}}(\Xi_0)$.

**Population Neural GTD.** We study a population neural GTD algorithm with:

$$\boldsymbol{\theta}_{k+1} = \mathcal{P}_{S_B}\Big\{\boldsymbol{\theta}_k - \beta_k\nabla_{\boldsymbol{\theta}}J_\upsilon(\boldsymbol{\theta}_k,\boldsymbol{w}_k)\Big\},\ \boldsymbol{w}_{k+1} = \mathcal{P}_{S_B}\Big\{\boldsymbol{w}_k + \beta_k\nabla_{\boldsymbol{w}}J_\upsilon(\boldsymbol{\theta}_k,\boldsymbol{w}_k)\Big\}, \tag{19}$$

where we assume that the *exact* gradient of the population objective function $J_\upsilon(\boldsymbol{\theta}_k,\boldsymbol{w}_k)$ is available. This form of update is relevant to a 'batch' neural GTD algorithm where samples of state transitions are collected a-priori. We have the following finite-time convergence result for (19):

---

**Theorem 3.1.** *Assume that H1, H2 hold, and the step size satisfies $\sup_{k\ge0}\beta_k \le \frac{1\wedge\upsilon}{8((1\vee\upsilon)^2+(1+\gamma)^2)}$. For the iterates generated by (19) and any $n \ge 1$, it holds:*

$$\min_{k\in\{0,...,n\}}\mathbb{E}_{\mathsf{init}}[d_{\widehat{f}}(\boldsymbol{z}_k,\widehat{\boldsymbol{z}}(\Xi_0))] \le \mathrm{C}_0^{\mathsf{p}}\left(\frac{B^3}{(1\wedge\upsilon)m^{\frac{1}{4}}}\right) + \mathrm{C}_1^{\mathsf{p}}\left(\frac{\|\boldsymbol{z}_0-\widehat{\boldsymbol{z}}\|_2^2}{(1\wedge\upsilon)\sum_{k=0}^n\beta_k}\right), \tag{20}$$

*for some constants $\mathrm{C}_0^{\mathsf{p}}, \mathrm{C}_1^{\mathsf{p}}$ that are independent of $m$, $B$, and $d_{\widehat{f}}(\boldsymbol{z}_k,\widehat{\boldsymbol{z}})$ is defined in (18). Moreover, the expectation $\mathbb{E}_{\mathsf{init}}[\cdot]$ is taken over the initialization of the NN $\Xi_0$.*

---

**Stochastic Neural GTD.** Next we focus on using stochastic samples in Algorithm 1. First, in the simplest setting, the state pairs $(s_k, s'_k)$ are drawn i.i.d. according to $s_k \sim \mu^\Pi$, $a_k \sim \Pi(\cdot|s_k)$, $s'_k \sim \mathrm{P}^{a_k}(s_k,\cdot)$, see line 5 of the pseudo code. We can rewrite the algorithm as

$$\boldsymbol{\theta}_{k+1} = \mathcal{P}_{S_B}\Big\{\boldsymbol{\theta}_k - \beta_k(\nabla_{\boldsymbol{\theta}}J_\upsilon(\boldsymbol{z}_k) + \boldsymbol{e}_k^{(1)})\Big\},\ \boldsymbol{w}_{k+1} = \mathcal{P}_{S_B}\Big\{\boldsymbol{w}_k + \beta_k(\nabla_{\boldsymbol{w}}J_\upsilon(\boldsymbol{z}_k) + \boldsymbol{e}_k^{(2)})\Big\}, \tag{21}$$

where $\boldsymbol{e}_k^{(i)}, i = 1, 2$ are the noise due to taking i.i.d. samples. Denote $\widetilde{s}_k := (s_k, a_k, s'_k)$, one has

$$\boldsymbol{e}_k^{(1)} = \nabla_{\boldsymbol{\theta}}J_\upsilon(\boldsymbol{\theta}_k,\boldsymbol{w}_k;\widetilde{s}_k) - \nabla_{\boldsymbol{\theta}}J_\upsilon(\boldsymbol{\theta}_k,\boldsymbol{w}_k), \boldsymbol{e}_k^{(2)} = \nabla_{\boldsymbol{w}}J_\upsilon(\boldsymbol{\theta}_k,\boldsymbol{w}_k;\widetilde{s}_k) - \nabla_{\boldsymbol{w}}J_\upsilon(\boldsymbol{\theta}_k,\boldsymbol{w}_k), \tag{22}$$

where $\nabla J_\upsilon(\boldsymbol{\theta}_k,\boldsymbol{w}_k;\widetilde{s}_k)$ denote the sampled gradients using the state pairs $\widetilde{s}_k$, see (13). Denote $\mathcal{F}_k$ as the filtration of the random variables $\{\boldsymbol{\theta}_0,\widetilde{s}_0,\widetilde{s}_1,...,\widetilde{s}_k\}$. Assume the following holds:

**H3.** *For any $k \ge 0$, there exists a constant $\sigma$ such that for any $i = 1, 2$,*

$$\mathbb{E}\big[\boldsymbol{e}_k^{(i)}|\mathcal{F}_{k-1}\big] = 0,\ \ \mathbb{E}\big[\|\boldsymbol{e}_k^{(i)}\|_2^2|\mathcal{F}_{k-1}\big] \le \sigma^2, \tag{23}$$

In other words, the noise vectors $e_k^{(i)}$, $i = 1, 2$ are martingale differences adapted to the filtration $(\mathcal{F}_k)_{k\geq 0}$. The second condition in (23) can be implied by the boundedness of $z_k$, in fact, using similar techniques as in Cai et al. [2019], it can be shown that $\sigma^2 = \mathcal{O}(B^2)$.

---

**Theorem 3.2.** *Assume H1, H2, H3 hold and the step size satisfies $\sup_{k\geq 0} \beta_k \leq \frac{1 \wedge \upsilon}{12((1 \vee \upsilon)^2 + (1+\gamma)^2)}$.* *For the iterates generated by Algorithm 1 and any $n \geq 1$, it holds:*

$$\mathbb{E}_{I_n, \text{init}}\left[d_{\widehat{f}}(z_{I_n}, \widehat{z}(\Xi_0))\right] \leq \mathrm{C}_0^{\mathsf{s}}\left(\frac{B^3}{(1 \wedge \upsilon)m^{\frac{1}{4}}}\right) + \mathrm{C}_1^{\mathsf{s}}\left(\frac{\|z_0 - \widehat{z}\|_2^2 + \sigma^2 \sum_{k=0}^n \beta_k^2}{(1 \wedge \upsilon)\sum_{k=0}^n \beta_k}\right), \quad (24)$$

*for some constants $\mathrm{C}_0^{\mathsf{s}}, \mathrm{C}_1^{\mathsf{s}}$ that are independent of $m$, $B$, and the function $d_{\widehat{f}}(z_{I_n}, \widehat{z}(\Xi_0))$ was defined in (18). The expectation above is taken over the independent r.v. $I_n$, the initialization to the NN $\Xi_0$, and the i.i.d. samples of states drawn from behavior policy during the algorithm.*

---

**Discussions of Theorem 3.1 and 3.2.** The conclusions (20), (24) imply that neural GTD finds a *saddle point* to the regularized MSBE problem with linearized NN function (15). For the population neural GTD, if we set $\beta_k$ to be constant, then the last term in (20) decays to zero at the rate $\mathcal{O}(1/n)$; for the stochastic neural GTD, setting a step size as $\beta_k = \mathcal{O}(1/\sqrt{k})$ and the last term in (24) decays at the rate of $\mathcal{O}(\log n/\sqrt{n})$. These rates are comparable to exact and stochastic primal-dual gradient methods, respectively, see [Chambolle and Pock, 2016, Juditsky et al., 2011]. Meanwhile, the first terms represent the bias controlled by the width of the 2-layer NN, and the bias is in the order of $\mathcal{O}(B^3 m^{-1/4})$. Importantly, we observe that the error bound converges to zero when $n, m \to \infty$.

For both theorems, the error bound $d_{\widehat{f}}(z_k, \widehat{z})$ computes the $L^2$ distance between the linearized NN functions, taken over behavior policy's stationary distribution $\mu^\Pi$. This optimality measure on the *function space* is used instead of the Euclidean norm $\|z_k - \widehat{z}\|_2^2$ of the *parameter space* so as to avoid trivial bounds since $z_k$ is $2md$-dimensional (and we consider $m \to \infty$).

**Markov Samples.** An alternative version of Theorem 3.2 is derived when Markov samples are used, i.e., the sampled state-pair $\widetilde{s}_k = (s_k, a_k, s_{k+1})$ is drawn from a single sample path of the Markov chain induced by the MDP and behavior policy $\Pi$, see line 4 & 7 of Algorithm 1.

---

**Theorem 3.3.** *Assume H1, the step size satisfies $|\beta_k - \beta_{k+1}| \leq \xi\beta_k^2$ for some constant $\xi$, $\sup_{k\geq 0} \beta_k \leq \frac{1 \wedge \upsilon}{12((1 \vee \upsilon)^2 + (1+\gamma)^2)}$, $\sup_{a,s} \rho(a|s) = \bar{\rho}$. Consider Algorithm 1 with Markov samples and any $n \geq 1$. With probability at least $1 - e^{\Omega(\log^2 m)}$ over NN initializations, it holds*

$$\mathbb{E}_{I_n}\left[d_{\widehat{f}}(z_{I_n}, \widehat{z})\right] = \mathcal{O}\left(\frac{B^{\frac{8}{3}}(\log m)^{\frac{3}{2}}/(1-\rho)}{(1 \wedge \upsilon)m^{\frac{1}{6}}} + \frac{\|z_0 - \widehat{z}\|_2^2 + \frac{B^2}{1-\rho} + (\frac{B^2}{1-\rho})\sum_{k=0}^n \beta_k^2}{(1 \wedge \upsilon)\sum_{k=0}^n \beta_k}\right), \quad (25)$$

*where $d_{\widehat{f}}(z_{I_n}, \widehat{z}(\Xi_0))$ was defined in (18) and $\rho \in [0, 1)$ is the convergence rate of the Markov chain. The expectation is taken over for r.v. $I_n$, and the sample path of Markov chain $(\widetilde{s}_0, \widetilde{s}_1, ...)$.*

---

Details are in Appendix D where we specify additional conditions on the Markov chain induced by the behavior policy. There are two differences from Theorem 3.2. First, the above holds in high probability w.r.t. the NN initialization. Second, the bias, variance are $\mathcal{O}(B^{\frac{8}{3}}(\log m)^{\frac{3}{2}}(1-\rho)^{-1}m^{-1/6})$, $\mathcal{O}(B^2(1-\rho)^{-1})$, which depends on the mixing time of Markov chain.

### 3.2 Minimizing the MSBE with Neural GTD Algorithms

Theorems 3.1 & 3.2 show the neural GTD algorithms converge to an optimal solution of (15) when $n, m \to \infty$. To show that an optimal solution of (15) is also optimal to (7), we consider:

**H4.** *For any $\theta \in S_B$, there exists $w(\theta) \in S_B$ such that $\mathbb{E}_{\text{init},s \sim \mu^\Pi}[|\bar{\delta}(s, \theta) - f(s, w(\theta))|^2] \leq \mathsf{c}_{\text{nn}}$, where $\mathsf{c}_{\text{nn}} \geq 0$ and the expectation is taken w.r.t. $s \sim \mu^\Pi$ and the NN initializations $\Xi_0$.*

The above assumption depends on the reward function $\mathrm{R}(s, a)$. Particularly, we have $\mathsf{c}_{\text{nn}} = 0$ if the TD error function lies in the function class of 2-layer ReLU NNs. Furthermore, we anticipate that $\mathsf{c}_{\text{nn}} \ll 1$ under the overparameterization setting $m \gg 1$. This is due to the representation power of such NNs as demonstrated in the recent works, e.g., [Neyshabur and Li, 2019].

Based on the above results, for any $\theta \in S_B$, we can control the MSBE $J_\upsilon(\theta) - J_\upsilon(\theta^\star)$ as

**Theorem 3.4.** *Assume H1,H2,H4, and the importance ratio is bounded as $\sup_{a,s} \rho(a|s) = \bar{\rho}$. Let $\boldsymbol{\theta}^{\star}(\Xi_0)$ be an optimal solution to (7). For any $\boldsymbol{\theta} \in S_B$,*

$$
\begin{aligned}
\mathbb{E}_{\mathsf{init}}\big[J_{\upsilon}(\boldsymbol{\theta}) - J_{\upsilon}(\boldsymbol{\theta}^{\star}(\Xi_0))\big] &\leq \mathrm{C}_0^J \mathbb{E}_{\mathsf{init}}\big[d_{\widehat{f}}(\boldsymbol{z}, \widehat{\boldsymbol{z}}(\Xi_0))\big] \\
&+ \mathrm{C}_1^J(B + B^{\frac{3}{2}}m^{-\frac{1}{4}})\sqrt{\mathbb{E}_{\mathsf{init}}\big[d_{\widehat{f}}(\boldsymbol{z}, \widehat{\boldsymbol{z}}(\Xi_0))\big]} + \mathcal{O}\big(B^3 m^{-1/2} + B^{5/2}m^{-1/4} + \mathsf{c}_{\mathsf{nn}}\big)
\end{aligned}
\tag{26}
$$

*for some constants $\mathrm{C}_0^J, \mathrm{C}_1^J$ that are independent of $B, m$. In the above, $\boldsymbol{z}$ is defined as the vector $\boldsymbol{z} = (\boldsymbol{\theta}, \boldsymbol{w})$ for any $\boldsymbol{w} \in S_B$, and $d_{\widehat{f}}(\boldsymbol{z}, \widehat{\boldsymbol{z}}(\Xi_0))$ was defined in (18).*

The difference $J_{\upsilon}(\boldsymbol{\theta}) - J_{\upsilon}(\boldsymbol{\theta}^{\star}(\Xi_0))$ corresponds to the sub-optimality of a given NN parameter $\boldsymbol{\theta}$ to the regularized mean squared Bellman error objective function. The above theorem quantifies this sub-optimality in terms of the distance to a saddle point of the problem (15).

Combining Theorem 3.4 with the previous analysis in Theorem 3.1 & 3.2, we obtain the *global convergence* guarantees of MSBE for the neural GTD algorithms as follows:

**Corollary 3.1.** *Assume H1,H2,H4 and the importance ratio is bounded as $\sup_{a,s} \rho(a|s) = \bar{\rho}$. We have the following guarantees for Neural GTD algorithms:*

- *Consider the population neural GTD algorithm [cf. (19)]. Set $\beta_k = \mathcal{O}(1)$. For any $n \geq 1$,*

$$
\begin{aligned}
\min_{k \in \{0,\dots,n\}} \mathbb{E}_{\mathsf{init}}\big[J_{\upsilon}(\boldsymbol{\theta}_k) - J_{\upsilon}(\boldsymbol{\theta}^{\star}(\Xi_0))\big] &= \mathcal{O}\big(n^{-1} + (B + B^{\frac{3}{2}}m^{-\frac{1}{4}})n^{-\frac{1}{2}}\big) \\
&+ \mathcal{O}\big(B^3 m^{-\frac{1}{4}} + B^{\frac{5}{2}}m^{-\frac{1}{8}} + \mathsf{c}_{\mathsf{nn}}\big),
\end{aligned}
\tag{27}
$$

- *Consider the stochastic neural GTD algorithm with i.i.d. samples [cf. Algorithm 1]. Assume in addition H3 and set $\beta_k = \mathcal{O}(1/\sqrt{k})$. For any $n \geq 1$,*

$$
\begin{aligned}
\mathbb{E}_{\mathsf{init},I_n}\big[J_{\upsilon}(\boldsymbol{\theta}_{I_n}) - J_{\upsilon}(\boldsymbol{\theta}^{\star}(\Xi_0))\big] &= \tilde{\mathcal{O}}\big(\sigma^2\, n^{-\frac{1}{2}} + (B + B^{\frac{3}{2}}m^{-\frac{1}{4}})\sigma\, n^{-\frac{1}{4}}\big) \\
&+ \mathcal{O}\big(B^3 m^{-\frac{1}{4}} + B^{\frac{5}{2}}m^{-\frac{1}{8}} + \mathsf{c}_{\mathsf{nn}}\big),
\end{aligned}
\tag{28}
$$

*where the $\tilde{\mathcal{O}}(\cdot)$ notation hides the logarithm terms in the upper bounds.*

*The expectations above are taken over the NN initialization $\Xi_0$ and the number of iterations $I_n$.*

When the bias term $\mathsf{c}_{\mathsf{nn}}$ is small, the above corollary shows that as $n, m \to \infty$, the neural GTD algorithms find an NN parameter $\boldsymbol{\theta}_{I_n}$ which *globally minimizes* the MSBE (7). Moreover, similar conclusions can be drawn for the Markov sample settings.

### 3.3 Preliminary Numerical Experiment

We perform preliminary experiments to support the above theories on a toy example of off-policy learning. We consider an MDP taken from the Garnet class with $|\mathsf{S}| = 500$ states, $|\mathsf{A}| = 5$ possible actions per state with uniformly distributed rewards, and the discount factor is $\gamma = 0.9$. We generate two random policies with the same support as the behavior/target policies, respectively. In Fig. 1, we compare the average MSBE against the number of neurons $m$, using a 2-layer, ReLU NN with random initialization according to H1, after $T = 3 \times 10^5$ iterations of neural GTD and neural TD [Cai et al., 2019] run with Markovian samples [cf. Algorithm 1], from 10 independent runs of state/action.

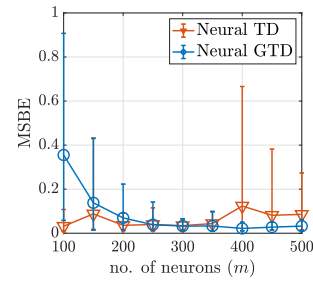

**Figure 1:** Comparing the averaged MSBE after 10 runs of Neural TD and Neural GTD in off-policy learning.

From the figure, we observe that the average MSBE (solid line) decreases with $m$ stably for neural GTD, as predicted by the above theorems. Meanwhile, the MSBE fluctuates with $m$ with neural TD, indicating that the latter can be unstable in the tested off-policy setting. Note that neural TD algorithm [Cai et al., 2019] has only been analyzed with on-policy data.

## 4 Proof Sketches

This section highlights the major steps involved in showing our main claims from the previous section. We first review on the approximation quality of the NN function, and develop its consequences in

the GTD learning paradigm. This will lead to our main theorems in Theorem 3.1 & 3.2. Then, we perform a perturbation analysis in light of $J_\upsilon(\boldsymbol{\theta}; \Xi_0)$ to yield Theorem 3.4.

**Approximating an NN function by Linearization.** To establish Theorem 3.1 & 3.2, we observe:

**Lemma 4.1.** *[Cai et al., 2019, Lemma 5.1] Under H1, H2, there exists constant $c_0$ where for any $\boldsymbol{\theta} \in S_B$,*

$$\mathbb{E}_{\mathsf{init}, s \sim \mu^\Pi} \left[ |f(s, \boldsymbol{\theta}) - \widehat{f}(s, \boldsymbol{\theta})|^2 \right] \leq c_0 B^3 \, m^{-1/2}. \tag{29}$$

Essentially, the expected approximation error of the function values $f(\cdot, \boldsymbol{\theta})$ by $\widehat{f}(\cdot, \boldsymbol{\theta})$ decays as $\mathcal{O}(m^{-1/2})$ for any $\boldsymbol{\theta} \in S_B$. As $m \to \infty$, the NN function behaves like a *linear function*.

**Neural GTD as Biased Gradient.** Our next step is to analyze the convergence of neural GTD learning [cf. (19) or Algorithm 1]. To this regard, we treat the algorithms as *biased primal-dual gradient* methods for (15), even though the updates have been designed for (10). Concretely, we consider the population neural GTD (19). Observe that

$$\nabla_{\boldsymbol{\theta}} J_\upsilon(\boldsymbol{\theta}_k, \boldsymbol{w}_k) = \nabla_{\boldsymbol{\theta}} \widehat{J}_\upsilon(\boldsymbol{\theta}_k, \boldsymbol{w}_k) + \widehat{\boldsymbol{e}}_k^{(1)}, \ \nabla_{\boldsymbol{w}} J_\upsilon(\boldsymbol{\theta}_k, \boldsymbol{w}_k) = \nabla_{\boldsymbol{w}} \widehat{J}_\upsilon(\boldsymbol{\theta}_k, \boldsymbol{w}_k) + \widehat{\boldsymbol{e}}_k^{(2)}, \tag{30}$$

where $\widehat{\boldsymbol{e}}_k^{(i)}, i = 1, 2$ represent the discrepancies between the gradient of $\widehat{J}_\upsilon(\boldsymbol{\theta}, \boldsymbol{w}), J_\upsilon(\boldsymbol{\theta}, \boldsymbol{w})$:

$$\widehat{\boldsymbol{e}}_k^{(1)} = \mathbb{E} \left[ f(s, \boldsymbol{w}_k) \nabla \bar{\delta}(s, \boldsymbol{\theta}_k) - \widehat{f}(s, \boldsymbol{w}_k) \nabla \widehat{\delta}(s, \boldsymbol{\theta}_k) + \upsilon \{ f(s, \boldsymbol{\theta}_k) \nabla f(s, \boldsymbol{\theta}_k) - \widehat{f}(s, \boldsymbol{\theta}_k) \nabla \widehat{f}(s, \boldsymbol{\theta}_k) \} \right]$$

$$\widehat{\boldsymbol{e}}_k^{(2)} = \mathbb{E} \left[ \left( \bar{\delta}(s, \boldsymbol{\theta}_k) - f(s, \boldsymbol{w}_k) \right) \nabla_{\boldsymbol{w}} f(s, \boldsymbol{w}_k) - \left( \widehat{\delta}(s, \boldsymbol{\theta}_k) - \widehat{f}(s, \boldsymbol{w}_k) \right) \nabla_{\boldsymbol{w}} \widehat{f}(s, \boldsymbol{w}_k) \right],$$

where the expectations are taken w.r.t. $s \sim \mu^\Pi$. We recall that $\widehat{\delta}(s, \boldsymbol{\theta}_k)$ is the TD error with the linearized NN function parameterized by $\boldsymbol{\theta}_k$. Therefore, each of the above terms represent differences between the NN function and its linear approximation.

As observed in Lemma 4.1, the above discrepancies in the function value diminishes as $m \to \infty$. This observation also extends to the associated gradients as we prove that

**Lemma 4.2.** *Under H1, H2, it holds for any $k \geq 0$ that*

$$\mathbb{E}_{\mathsf{init}}[\|\widehat{\boldsymbol{e}}_k^{(1)}\|_2^2] \vee \mathbb{E}_{\mathsf{init}}[\|\widehat{\boldsymbol{e}}_k^{(2)}\|_2^2] \leq \mathrm{C}_0 B^3 m^{-1/2}, \tag{31}$$

*for some constant $\mathrm{C}_0$ that is independent of $B, m$, the above expectations are taken with respect to the initialization parameters $\Xi_0$ of the NN.*

From Lemma 4.2, it is clear that if $m$ is sufficiently large, then the population neural GTD algorithm (19) follow closely a primal-dual gradient method for the *convex-concave* problem (15).

If we assume that $\widehat{\boldsymbol{e}}_k^{(1)} = \widehat{\boldsymbol{e}}_k^{(2)} = 0$, then the convergence of (19) to a global optimal solution is guaranteed by the classical analysis from, e.g., [Chambolle and Pock, 2016, He and Yuan, 2012]. Fix $n \in \mathbb{N}$, the results from [Chambolle and Pock, 2016] shows that using a slight modification to (19), one can find an $\mathcal{O}(1/n)$ saddle point $(\overline{\boldsymbol{\theta}}_n, \overline{\boldsymbol{w}}_n)$ in $n$ iterations. However, such result does not immediately relate to a bound on the MSBE $J_\upsilon(\boldsymbol{\theta}_n)$ which will be needed later. In addition, the analysis in [Chambolle and Pock, 2016] does not consider bias in the gradient (30). Lastly, although the objective function $\widehat{J}_\upsilon(\boldsymbol{\theta}, \boldsymbol{w})$ admits a quadratic form, we note that the parameters $\boldsymbol{\theta}, \boldsymbol{w}$ are both $md$-dimensional vectors. As $m \to \infty$, the strong convexity/concavity modulus in the Euclidean space associated with the objective function may approach zero. Also see [Lorenz and Pock, 2015] for extensions of the primal-dual algorithm to Hilbert space.

In light of the above challenges, we adopt an error bound metric that adapts to the problem structure at hand. A natural choice is the $L^2$ distance between $\widehat{f}(\cdot, \boldsymbol{\theta}), \widehat{f}(\cdot, \widehat{\boldsymbol{\theta}})$ in the *function space*, as defined in $d_{\widehat{f}}(\boldsymbol{z}, \widehat{\boldsymbol{z}})$ [cf. (18)]. Define the primal-dual gradient operator on $\boldsymbol{z} = (\boldsymbol{\theta}, \boldsymbol{w})$ as:

$$\widehat{\Phi}(\boldsymbol{z}) := \left( \nabla_{\boldsymbol{\theta}} \widehat{J}_\upsilon(\boldsymbol{z})^\top \ - \nabla_{\boldsymbol{w}} \widehat{J}_\upsilon(\boldsymbol{z})^\top \right)^\top. \tag{32}$$

**Lemma 4.3.** *Let $\widehat{\boldsymbol{z}}$ be a saddle point of the problem (15). For any $\boldsymbol{z} = (\boldsymbol{\theta}, \boldsymbol{w}) \in S_B \times S_B$. Set $\mu = \min\{1, \upsilon\}$ and $L_\Phi = 4((\min\{1, \upsilon\})^2 + (1 + \gamma)^2)$, it holds*

$$\langle \widehat{\Phi}(\boldsymbol{z}) - \widehat{\Phi}(\widehat{\boldsymbol{z}}), \boldsymbol{z} - \boldsymbol{z}^\star \rangle \geq \mu \, d_{\widehat{f}}(\boldsymbol{z}, \widehat{\boldsymbol{z}}), \ \ \|\widehat{\Phi}(\boldsymbol{z}) - \widehat{\Phi}(\widehat{\boldsymbol{z}})\|_2^2 \leq L_\Phi \, d_{\widehat{f}}(\boldsymbol{z}, \widehat{\boldsymbol{z}}). \tag{33}$$

Note that the constants $\mu, L_\Phi$ are *independent* of the problem dimension $m$. The condition (33) shows that the primal-dual gradient operator $\widehat{\Phi}(z)$ is a smooth monotone operator.

Lemma 4.2 & 4.3 show that the population neural GTD algorithm is a *biased* primal-dual gradient method on the convex-concave saddle point problem (15). In the appendix, we will show

$$\mathbb{E}[\|\boldsymbol{z}_{k+1} - \widehat{\boldsymbol{z}}\|_2^2] \leq \mathbb{E}\big[\|\boldsymbol{z}_k - \widehat{\boldsymbol{z}}\|_2^2 - 2\mu\beta_k d_{\widehat{f}}(\boldsymbol{z}_k, \widehat{\boldsymbol{z}}) + \beta_k^2 L_\Phi d_{\widehat{f}}(\boldsymbol{z}_k, \widehat{\boldsymbol{z}})\big] + \mathcal{O}\big(\beta_k B^3 / m^{\frac{1}{4}}\big), \quad (34)$$

where the expectation is taken with respect to the NN initialization. Taking a summation of the above inequalities from $k = 0$ to $k = n$ and canceling terms yield Theorem 3.1.

In the i.i.d. sample case, the stochastic neural GTD algorithm can be analyzed in a similar manner through exploiting the conditional zero mean and bounded variance properties in H3. The latter leads to Theorem 3.2. In the Markov sample case, we utilize the Poisson equation which decomposes the noise terms into a martingale part and a Markov part. We show that the Markov part is small in magnitude. The derivations are similar to [Karimi et al., 2019], yet we have adapted the analysis to the primal-dual gradient method. In addition, inspired by [Gao et al., 2019], we derive similar bounds to Lemma 4.2 which hold in high probability over the initialization. This leads to Theorem 3.3, see Appendix D for the details.

**Finding Global Minimizer of MSBE.** Our last task is to evaluate the solution quality of the output from neural GTD in terms of deviation from the minimum regularized MSBE (7).

To derive Theorem 3.4, we shall exploit H4 and the Fenchel's conjugation in (8). In particular, it can be shown that

$$J_\upsilon(\boldsymbol{\theta}) \overset{(a)}{\leq} J_\upsilon(\widehat{\boldsymbol{\theta}}) + \mathcal{O}\big(m^{-1/4} + d_{\widehat{f}}(\boldsymbol{z}, \widehat{\boldsymbol{z}})^{\frac{1}{2}}\big) \overset{(b)}{\leq} J_\upsilon(\widehat{\boldsymbol{\theta}}, \boldsymbol{w}(\widehat{\boldsymbol{\theta}})) + \mathcal{O}\big(m^{-1/4} + d_{\widehat{f}}(\boldsymbol{z}, \widehat{\boldsymbol{z}})^{\frac{1}{2}} + \mathsf{c}_{\mathsf{nn}}\big)$$

where (a) can be derived using Lemma 4.1, and (b) is due to H4 which guarantees the existence of $\boldsymbol{w}(\widehat{\boldsymbol{\theta}})$. Note that $J_\upsilon(\widehat{\boldsymbol{\theta}}, \boldsymbol{w}(\widehat{\boldsymbol{\theta}}))$ is the primal-dual objective function defined in (10). Subsequently, using Lemma 4.1, we obtain

$$J_\upsilon(\widehat{\boldsymbol{\theta}}, \boldsymbol{w}(\widehat{\boldsymbol{\theta}})) \leq \widehat{J}_\upsilon(\widehat{\boldsymbol{\theta}}, \boldsymbol{w}(\widehat{\boldsymbol{\theta}})) + \mathcal{O}(m^{-1/4}) \leq \widehat{J}_\upsilon(\boldsymbol{\theta}^\star, \widehat{\boldsymbol{w}}) + \mathcal{O}(m^{-1/4}) \quad (35)$$

where the last inequality is due to $\widehat{J}_\upsilon(\widehat{\boldsymbol{\theta}}, \boldsymbol{w}(\widehat{\boldsymbol{\theta}})) \leq \widehat{J}_\upsilon(\widehat{\boldsymbol{\theta}}, \widehat{\boldsymbol{w}})$ and the fact $(\widehat{\boldsymbol{\theta}}, \widehat{\boldsymbol{w}})$ is a saddle point to (15). Applying Lemma 4.1 again yields the inequality

$$\widehat{J}_\upsilon(\boldsymbol{\theta}^\star, \widehat{\boldsymbol{w}}) \leq J_\upsilon(\boldsymbol{\theta}^\star, \widehat{\boldsymbol{w}}) + \mathcal{O}(m^{-1/4}) \leq J_\upsilon(\boldsymbol{\theta}^\star) + \mathcal{O}(m^{-1/4}), \quad (36)$$

where the last inequality is due to the optimality of $\bar{\delta}(\cdot; \boldsymbol{\theta}^\star)$ for the maximization in (8). We remark that the above inequalities hold in an expectation taken over the initialization $\Xi_0$ of NN [cf. H1].

Collecting terms in the above leads to Theorem 3.4 which shows

$$J_\upsilon(\boldsymbol{\theta}) - J_\upsilon(\boldsymbol{\theta}^\star) = \mathcal{O}(m^{-1/4} + \mathsf{c}_{\mathsf{nn}} + d_{\widehat{f}}(\boldsymbol{z}, \widehat{\boldsymbol{z}})^{\frac{1}{2}}). \quad (37)$$

Combined with Theorem 3.1 & 3.2, we conclude that the neural GTD algorithms find a global minimizer to the regularized MSBE problem (7), i.e., justifying our main claims in Corollary 3.1.

## 5 Conclusions

We have derived the first neural GTD learning algorithm for off-policy learning and proved its *global convergence* to a minimizer of the regularized MSBE. The main idea is to use a Fenchel conjugate's equivalent formulation to the MSBE objective function and design a novel objective function that involves two NNs. We consider different sample requirements (population, i.i.d. samples and Markov samples), and analyze the convergence rates to a global MSBE minimizer.

**Acknowledgement & Funding Disclosure**  The authors would like to thank Mr. Alan Lun (CUHK) for conducting the preliminary numerical experiments in this paper. H.-T. Wai is supported by the CUHK Direct Grant #4055113. M. Hong is supported in part by NSF under Grant CCF-1651825, CMMI-172775, CIF-1910385 and by AFOSR under grant 19RT0424.

**Broader Impact**  This work does not present any foreseeable societal consequence.

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
