[Supplementary Material]

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

where $s'$ denotes the next state that is transitioned into. The function $\mathrm{R}(s,a)$ is the reward received after action $a$ in state $s$; lastly, $\gamma \in (0,1)$ is the discount factor. We assume $\sup_{s,a} \mathrm{R}(s,a) \leq \overline{r}$.

The *target policy* $\pi(a|s)$ is the conditional probability of action $a \in \mathsf{A}$ given the current state $s \in \mathsf{S}$ [Szepesvári, 2010]. This induces a Markov chain with the transition kernel $\mathrm{P}^\pi(s,\cdot) := \mathbb{E}_{a\sim\pi(\cdot|s)}[\mathrm{P}^a(s,\cdot)]$. The *policy evaluation* problem aims at learning a *value function* $\mathrm{V}^\pi : \mathsf{S} \to \mathbb{R}$, defined as the discounted total reward starting from state $s$:

$$\mathrm{V}^\pi(s) := \mathbb{E}\big[ \textstyle\sum_{t=0}^\infty \gamma^t \mathrm{R}(s_t, a_t) \,\big|\, s_0 = s,\ a_t \sim \pi(\cdot|s_t),\ s_{t+1} \sim \mathrm{P}^{a_t}(s_t, \cdot)\big]. \qquad (2)$$

Applying the Bellman equation [Van Hasselt, 2012] shows that

$$0 = \mathrm{V}^\pi(s) - \mathbb{E}_{a\sim\pi(\cdot|s)}\big[\mathrm{R}(s,a)\big] - \gamma\mathrm{P}^\pi\mathrm{V}^\pi(s). \qquad (3)$$

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

}[\|z_{k+1} - \widehat{z}\|_2^2] \leq \mathbb{E}\big[\|z_k - \widehat{z}\|_2^2 - 2\mu\beta_k d_{\widehat{f}}(z_k, \widehat{z}) + \beta_k^2 L_\Phi d_{\widehat{f}}(z_k, \widehat{z})\big] + \mathcal{O}\big(\beta_k B^3/m^{\frac{1}{4}}\big), \quad (34)$$

where the expectation is taken with respect to the NN initialization. Taking a summation of the above inequalities from $k = 0$ to $k = n$ and canceling terms yield Theorem 3.1.

In the i.i.d. sample case, the stochastic neural GTD algorithm can be analyzed in a similar manner through exploiting the conditional zero mean and bounded variance properties in H3. The latter leads to Theorem 3.2. In the Markov sample case, we utilize the Poisson equation which decomposes the noise terms into a martingale part and a Markov part. We show that the Markov part is small in magnitude. The derivations are similar to [Karimi et al., 2019], yet we have adapted the analysis to the primal-dual gradient method. In addition, inspired by [Gao et al., 2019], we derive similar bounds to Lemma 4.2 which hold in high probability over the initialization. This leads to Theorem 3.3, see Appendix D for the details.

**Finding Global Minimizer of MSBE.** Our last task is to evaluate the solution quality of the output from neural GTD in terms of deviation from the minimum regularized MSBE (7).

To derive Theorem 3.4, we shall exploit H4 and the Fenchel's conjugation in (8). In particular, it can be shown that

$$J_\upsilon(\theta) \overset{(a)}{\leq} J_\upsilon(\widehat{\theta}) + \mathcal{O}\big(m^{-1/4} + d_{\widehat{f}}(z, \widehat{z})^{\frac{1}{2}}\big) \overset{(b)}{\leq} J_\upsilon(\widehat{\theta}, w(\widehat{\theta})) + \mathcal{O}\big(m^{-1/4} + d_{\widehat{f}}(z, \widehat{z})^{\frac{1}{2}} + \mathsf{c_{nn}}\big)$$

where (a) can be derived using Lemma 4.1, and (b) is due to H4 which guarantees the existence of $w(\widehat{\theta})$. Note that $J_\upsilon(\widehat{\theta}, w(\widehat{\theta}))$ is the primal-dual objective function defined in (10). Subsequently, using Lemma 4.1, we obtain

$$J_\upsilon(\widehat{\theta}, w(\widehat{\theta})) \leq \widehat{J}_\upsilon(\widehat{\theta}, w(\widehat{\theta})) + \mathcal{O}(m^{-1/4}) \leq \widehat{J}_\upsilon(\theta^\star, \widehat{w}) + \mathcal{O}(m^{-1/4}) \quad (35)$$

where the last inequality is due to $\widehat{J}_\upsilon(\widehat{\theta}, w(\widehat{\theta})) \leq \widehat{J}_\upsilon(\widehat{\theta}, \widehat{w})$ and the fact $(\widehat{\theta}, \widehat{w})$ is a saddle point to (15). Applying Lemma 4.1 again yields the inequality

$$\widehat{J}_\upsilon(\theta^\star, \widehat{w}) \leq J_\upsilon(\theta^\star, \widehat{w}) + \mathcal{O}(m^{-1/4}) \leq J_\upsilon(\theta^\star) + \mathcal{O}(m^{-1/4}), \quad (36)$$

where the last inequality is due to the optimality of $\bar{\delta}(\cdot; \theta^\star)$ for the maximization in (8). We remark that the above inequalities hold in an expectation taken over the initialization $\Xi_0$ of NN [cf. H1].

Collecting terms in the above leads to Theorem 3.4 which shows

$$J_\upsilon(\theta) - J_\upsilon(\theta^\star) = \mathcal{O}(m^{-1/4} + \mathsf{c_{nn}} + d_{\widehat{f}}(z, \widehat{z})^{\frac{1}{2}}). \quad (37)$$

Combined with Theorem 3.1 & 3.2, we conclude that the neural GTD algorithms find a global minimizer to the regularized MSBE problem (7), i.e., justifying our main claims in Corollary 3.1.

## 5 Conclusions

We have derived the first neural GTD learning algorithm for off-policy learning and proved its *global convergence* to a minimizer of the regularized MSBE. The main idea is to use a Fenchel conjugate's equivalent formulation to the MSBE objective function and design a novel objective function that involves two NNs. We consider different sample requirements (population, i.i.d. samples and Markov samples), and analyze the convergence rates to a global MSBE minimizer.

**Acknowledgement & Funding Disclosure**    The authors would like to thank Mr. Alan Lun (CUHK) for conducting the preliminary numerical experiments in this paper. H.-T. Wai is supported by the CUHK Direct Grant #4055113. M. Hong is supported in part by NSF under Grant CCF-1651825, CMMI-172775, CIF-1910385 and by AFOSR under grant 19RT0424.

**Broader Impact**    This work does not present any foreseeable societal consequence.

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

# A  Proof of Theorem 3.1

We shall use the notations $(a \vee b) = \max\{a, b\}$ and $(a \wedge b) = \min\{a, b\}$.

Recall that (15) is given by:

$$\min_{\boldsymbol{\theta} \in S_B} \max_{\boldsymbol{w} \in S_B} \widehat{J}_\upsilon(\boldsymbol{\theta}, \boldsymbol{w}) := \mathbb{E}_{s \sim \mu^\Pi}\big[\widehat{f}(s, \boldsymbol{w})\widehat{\delta}(s, \boldsymbol{\theta})\big] - \frac{1}{2}\mathbb{E}_{s \sim \mu^\Pi}[\widehat{f}(s, \boldsymbol{w})^2] + \frac{\upsilon}{2}\mathbb{E}_{s \sim \mu^\Pi}[\widehat{f}(s, \boldsymbol{\theta})^2]. \quad (38)$$

To begin our analysis, denote $\boldsymbol{z}_k := (\boldsymbol{\theta}_k, \boldsymbol{w}_k)$ and the optimal primal-dual solution to (15) (or (38)) as $\widehat{\boldsymbol{z}} = (\widehat{\boldsymbol{\theta}}, \widehat{\boldsymbol{w}})$. In this regard, we can write the population update as:

$$\boldsymbol{\theta}_{k+1} = \mathcal{P}_{S_B}\Big\{\boldsymbol{\theta}_k - \beta_k\Big(\nabla_{\boldsymbol{\theta}}\widehat{J}_\upsilon(\boldsymbol{z}_k) + \widehat{\boldsymbol{e}}_k^{(1)}\Big)\Big\}, \quad \boldsymbol{w}_{k+1} = \mathcal{P}_{S_B}\Big\{\boldsymbol{w}_k + \alpha_k\Big(\nabla_{\boldsymbol{w}}\widehat{J}_\upsilon(\boldsymbol{z}_k) + \widehat{\boldsymbol{e}}_k^{(2)}\Big)\Big\}, \quad (39)$$

with the errors defined as:

$$\widehat{\boldsymbol{e}}_k^{(1)} = \mathbb{E}_{s \sim \mu^\Pi}\Big[f(s, \boldsymbol{w}_k)\nabla\bar{\delta}(s, \boldsymbol{\theta}_k) + \upsilon f(s, \boldsymbol{\theta}_k)\nabla f(s, \boldsymbol{\theta}_k) - \big(\widehat{f}(s, \boldsymbol{w}_k)\nabla\widehat{\delta}(s, \boldsymbol{\theta}_k) + \upsilon\widehat{f}(s, \boldsymbol{\theta}_k)\nabla\widehat{f}(s, \boldsymbol{\theta}_k)\big)\Big]$$

$$\widehat{\boldsymbol{e}}_k^{(2)} = \mathbb{E}_{s \sim \mu^\Pi}\Big[\big(\bar{\delta}(s, \boldsymbol{\theta}_k) - f(s, \boldsymbol{w}_k)\big)\nabla_{\boldsymbol{w}}f(s, \boldsymbol{w}_k) - \big((\widehat{\delta}(s, \boldsymbol{\theta}_k) - \widehat{f}(s, \boldsymbol{w}_k))\nabla_{\boldsymbol{w}}\widehat{f}(s, \boldsymbol{w}_k)\big)\Big], \quad (40)$$

whose magnitude is controlled by the following lemma:

**Lemma A.1.** *Under H1, H2, it holds that for any $k \geq 0$, we have*

$$\mathbb{E}_{\mathsf{init}}[\|\widehat{\boldsymbol{e}}_k^{(1)}\|_2^2] \vee \mathbb{E}_{\mathsf{init}}[\|\widehat{\boldsymbol{e}}_k^{(2)}\|_2^2] \leq \mathrm{C}_0 B^3 m^{-1/2}, \quad (41)$$

*for some constant $\mathrm{C}_0$ that is independent of $B, m, k$, the above expectations are taken with respect to the initialization of the NN.*

Define the concatenated gradient vectors as $\widehat{\Phi}_k(\boldsymbol{z}_k) = \big(\nabla_{\boldsymbol{\theta}}\widehat{J}_\upsilon(\boldsymbol{z}_k)^\top \ -\nabla_{\boldsymbol{w}}\widehat{J}_\upsilon(\boldsymbol{z}_k)^\top\big)^\top$. Using the fact that $\widehat{\boldsymbol{\theta}} = \mathcal{P}_{S_B}\{\widehat{\boldsymbol{\theta}} - \beta\nabla_{\boldsymbol{\theta}}\widehat{J}_\upsilon(\widehat{\boldsymbol{z}})\}$ for any $\beta \geq 0$ and the non-expansive property of projection, we observe that

$$\begin{aligned}\|\boldsymbol{\theta}_{k+1} - \widehat{\boldsymbol{\theta}}\|_2^2 &\leq \|\boldsymbol{\theta}_k - \beta_k(\nabla_{\boldsymbol{\theta}}\widehat{J}_\upsilon(\boldsymbol{z}_k) + \widehat{\boldsymbol{e}}_k^{(1)} - \nabla_{\boldsymbol{\theta}}\widehat{J}_\upsilon(\widehat{\boldsymbol{z}})) - \widehat{\boldsymbol{\theta}}\|_2^2 \\ &= \|\boldsymbol{\theta}_k - \widehat{\boldsymbol{\theta}}\|_2^2 + \beta_k^2\|\nabla_{\boldsymbol{\theta}}\widehat{J}_\upsilon(\boldsymbol{z}_k) + \widehat{\boldsymbol{e}}_k^{(1)} - \nabla_{\boldsymbol{\theta}}\widehat{J}_\upsilon(\widehat{\boldsymbol{z}})\|_2^2 \\ &\quad - 2\beta_k\langle\boldsymbol{\theta}_k - \widehat{\boldsymbol{\theta}}, \nabla_{\boldsymbol{\theta}}\widehat{J}_\upsilon(\boldsymbol{z}_k) + \widehat{\boldsymbol{e}}_k^{(1)} - \nabla_{\boldsymbol{\theta}}\widehat{J}_\upsilon(\widehat{\boldsymbol{z}})\rangle.\end{aligned} \quad (42)$$

Similarly, using $\widehat{\boldsymbol{w}} = \mathcal{P}_{S_B}\{\widehat{\boldsymbol{w}} - \alpha\nabla_{\boldsymbol{w}}\widehat{J}_\upsilon(\widehat{\boldsymbol{z}})\}$ for any $\alpha \geq 0$, we get

$$\begin{aligned}\|\boldsymbol{w}_{k+1} - \widehat{\boldsymbol{w}}\|_2^2 &\leq \|\boldsymbol{w}_k + \alpha_k(\nabla_{\boldsymbol{w}}\widehat{J}_\upsilon(\boldsymbol{z}_k) + \widehat{\boldsymbol{e}}_k^{(2)} - \nabla_{\boldsymbol{w}}\widehat{J}_\upsilon(\widehat{\boldsymbol{z}})) - \widehat{\boldsymbol{w}}\|_2^2 \\ &= \|\boldsymbol{w}_k - \widehat{\boldsymbol{w}}\|_2^2 + \beta_k^2\|\nabla_{\boldsymbol{w}}\widehat{J}_\upsilon(\boldsymbol{z}_k) + \widehat{\boldsymbol{e}}_k^{(2)} - \nabla_{\boldsymbol{w}}\widehat{J}_\upsilon(\widehat{\boldsymbol{z}})\|_2^2 \\ &\quad + 2\beta_k\langle\boldsymbol{w}_k - \widehat{\boldsymbol{w}}, \nabla_{\boldsymbol{w}}\widehat{J}_\upsilon(\boldsymbol{z}_k) + \widehat{\boldsymbol{e}}_k^{(2)} - \nabla_{\boldsymbol{w}}\widehat{J}_\upsilon(\widehat{\boldsymbol{z}})\rangle.\end{aligned} \quad (43)$$

Adding up the two inequalities lead to: for any $c_1 > 0$,

$$\begin{aligned}\|\boldsymbol{z}_{k+1} - \widehat{\boldsymbol{z}}\|_2^2 &\leq \|\boldsymbol{z}_k - \widehat{\boldsymbol{z}}\|_2^2 - 2\beta_k\langle\widehat{\Phi}(\boldsymbol{z}_k) - \widehat{\Phi}(\widehat{\boldsymbol{z}}), \boldsymbol{z}_k - \boldsymbol{z}^\star\rangle + \frac{\beta_k}{c_1}\|\boldsymbol{z}_k - \widehat{\boldsymbol{z}}\|_2^2 \\ &\quad + 2\beta_k^2\|\nabla\widehat{J}_\upsilon(\boldsymbol{z}_k) - \nabla\widehat{J}_\upsilon(\widehat{\boldsymbol{z}})\|_2^2 + \big(2\beta_k^2 + c_1\beta_k\big)\big\{\|\widehat{\boldsymbol{e}}_k^{(1)}\|_2^2 + \|\widehat{\boldsymbol{e}}_k^{(2)}\|_2^2\big\},\end{aligned} \quad (44)$$

where we have denoted $\nabla\widehat{J}_\upsilon(\boldsymbol{z}) = (\nabla_{\boldsymbol{\theta}}\widehat{J}_\upsilon(\boldsymbol{z})^\top \ \nabla_{\boldsymbol{w}}\widehat{J}_\upsilon(\boldsymbol{z})^\top)^\top$. Recall that $\widehat{\boldsymbol{z}} = (\widehat{\boldsymbol{\theta}}, \widehat{\boldsymbol{w}})$ is the saddle point solution to (38), we observe the following lemma (which appears in the main paper as Lemma 4.3) to be proven in Appendix A.2.

**Lemma A.2.** *There exist constants $\mu, L_\Phi$ such that it holds*

$$\frac{1}{\mu}\langle\widehat{\Phi}(\boldsymbol{z}_k) - \widehat{\Phi}(\widehat{\boldsymbol{z}}), \boldsymbol{z}_k - \widehat{\boldsymbol{z}}\rangle \geq \mathbb{E}_{s \sim \mu^\Pi}\big[|\widehat{f}(s, \boldsymbol{w}_k) - \widehat{f}(s, \widehat{\boldsymbol{w}})|^2 + |\widehat{f}(s, \boldsymbol{\theta}_k) - \widehat{f}(s, \widehat{\boldsymbol{\theta}})|^2\big] =: d_{\widehat{f}}(\boldsymbol{z}_k, \widehat{\boldsymbol{z}}),$$

$$\|\nabla\widehat{J}_\upsilon(\boldsymbol{z}_k) - \nabla\widehat{J}_\upsilon(\widehat{\boldsymbol{z}})\|_2^2 \leq L_\Phi\, d_{\widehat{f}}(\boldsymbol{z}_k, \widehat{\boldsymbol{z}}), \quad (45)$$

*for any $k \geq 0$, where $\mu = \min\{1, \upsilon\} = 1 \wedge \upsilon$ and $L_\Phi = 4((1 \vee \upsilon)^2 + (1 + \gamma)^2)$,*

The above lemma yields:

$$\|\boldsymbol{z}_{k+1} - \widehat{\boldsymbol{z}}\|_2^2 \leq \|\boldsymbol{z}_k - \widehat{\boldsymbol{z}}\|_2^2 - 2\beta_k\mu\,d_{\widehat{f}}(\boldsymbol{z}_k,\widehat{\boldsymbol{z}}) + \frac{\beta_k}{c_1}\|\boldsymbol{z}_k - \widehat{\boldsymbol{z}}\|_2^2$$
$$+ 2\beta_k^2 L_\Phi d_{\widehat{f}}(\boldsymbol{z}_k,\widehat{\boldsymbol{z}}) + \left(2\beta_k^2 + c_1\beta_k\right)\left\{\|\widehat{\boldsymbol{e}}_k^{(1)}\|_2^2 + \|\widehat{\boldsymbol{e}}_k^{(2)}\|_2^2\right\}. \tag{46}$$

Notice that $\|\boldsymbol{z}_k - \widehat{\boldsymbol{z}}\|_2^2 \leq 4B^2$, we obtain

$$2\beta_k\big(\mu - \beta_k L_\Phi\big)\,d_{\widehat{f}}(\boldsymbol{z}_k,\widehat{\boldsymbol{z}}) \leq \|\boldsymbol{z}_k - \widehat{\boldsymbol{z}}\|_2^2 - \|\boldsymbol{z}_{k+1} - \widehat{\boldsymbol{z}}\|_2^2 + 4B^2\frac{\beta_k}{c_1} + \left(2\beta_k^2 + c_1\beta_k\right)\left\{\|\widehat{\boldsymbol{e}}_k^{(1)}\|_2^2 + \|\widehat{\boldsymbol{e}}_k^{(2)}\|_2^2\right\}. \tag{47}$$

Note that we have chosen $\beta_k$ such that $\mu - \beta_k L_\Phi \geq \mu/2$. Setting $c_1 = m^{\frac{1}{4}}$ and using Lemma A.1, we obtain

$$\mu\,\beta_k\,d_{\widehat{f}}(\boldsymbol{z}_k,\widehat{\boldsymbol{z}}) \leq \|\boldsymbol{z}_k - \widehat{\boldsymbol{z}}\|_2^2 - \|\boldsymbol{z}_{k+1} - \widehat{\boldsymbol{z}}\|_2^2 + 4\beta_k^2\big(\frac{\mathrm{C}_0 B^3}{\sqrt{m}}\big) + 2\frac{\beta_k}{m^{\frac{1}{4}}}\big(2B^2 + \mathrm{C}_0 B^3\big). \tag{48}$$

Let $n \geq 1$, summing up both sides of the inequality from $k = 0$ to $k = n$ leads to

$$\min_{k \in \{0,\ldots,n\}} d_{\widehat{f}}(\boldsymbol{z}_k,\widehat{\boldsymbol{z}}) \leq \frac{1}{(1 \wedge \upsilon)\sum_{k=0}^n \beta_k} \left\{\|\boldsymbol{z}_0 - \widehat{\boldsymbol{z}}\|_2^2 + (4\mathrm{C}_0 B^3/\sqrt{m})\sum_{k=0}^n \beta_k^2\right\} + \frac{4B^2 + 2\mathrm{C}_0 B^3}{(1 \wedge \upsilon)m^{\frac{1}{4}}}. \tag{49}$$

Simplifying the constants conclude the proof.

## A.1 Proof of Lemma A.1

In this subsection, we should use the shorthand notation $\mathbb{E}_\mu[\cdot]$ to denote that the expectation is taken w.r.t. $s \sim \mu^\Pi$. Let us begin by borrowing the following lemmas from [Cai et al., 2019]

**Lemma A.3.** *[Cai et al., 2019, Lemma F.1] Under H1, H2, there exists constant $c_1$ such that it holds for any $\boldsymbol{\theta} \in S_B$ that*

$$\mathbb{E}_{\mathsf{init},\mu}\left[\frac{1}{m}\sum_{r=1}^m \mathbb{1}\{|\langle\boldsymbol{\theta}_0^{(r)}, \boldsymbol{x}_s\rangle| \leq \|\boldsymbol{\theta}^{(r)} - \boldsymbol{\theta}_0^{(r)}\|_2\}\right] \leq c_1 B\, m^{-1/2}. \tag{50}$$

The proof of the following lemma can be adapted from [Cai et al., 2019, Lemma F.2], see Appendix A.3 for the proof.

**Lemma A.4.** *Under H1, H2, there exists constant $c_2$ such that it holds for any $\boldsymbol{\theta} \in S_B$ that*

$$\mathbb{E}_{\mathsf{init},\mu}\left[|\widehat{f}(s, \boldsymbol{\theta}_0)|^2 \frac{1}{m}\sum_{r=1}^m \mathbb{1}\{|\langle\boldsymbol{\theta}_0^{(r)}, \boldsymbol{x}_s\rangle| \leq \|\boldsymbol{\theta}^{(r)} - \boldsymbol{\theta}_0^{(r)}\|_2\}\right] \leq c_2 B\, m^{-1/2} + m^{-1}. \tag{51}$$

Now, we begin proving the Lemma A.1. First we define a constant $\widetilde{\mathrm{C}}_0$ stated in the lemma as:

$$\widetilde{\mathrm{C}}_0 := \max\left\{\left\{8(1+\gamma^2)+4\upsilon^2\right\}\left\{c_0 B^3 + 2c_1 B^3 + 2c_2 B + 2\right\}, 6c_0 B^3 + 8(c_1\bar{r}^2 B + 3c_1 B^3 + 3c_2 B + 3)\right\}. \tag{52}$$

Based on the above lemmas, let us observe

$$\|\widehat{\boldsymbol{e}}_k^{(1)}\|_2^2 \leq 2\,\mathbb{E}_\mu\big[\underbrace{\|f(s, \boldsymbol{w}_k)\nabla\bar{\delta}(s, \boldsymbol{\theta}_k) - \widehat{f}(s, \boldsymbol{w}_k)\nabla\widehat{\delta}(s, \boldsymbol{\theta}_k)\|_2^2}_{=:E_k^{(1,1)}}\big]$$
$$+ 2\upsilon^2\,\mathbb{E}_\mu\big[\underbrace{\|f(s, \boldsymbol{\theta}_k)\nabla f(s, \boldsymbol{\theta}_k) - \widehat{f}(s, \boldsymbol{\theta}_k)\nabla\widehat{f}(s, \boldsymbol{\theta}_k)\|_2^2}_{=:E_k^{(1,2)}}\big]. \tag{53}$$

We have

$$E_k^{(1,1)} \leq 2|f(s, \boldsymbol{w}_k) - \widehat{f}(s, \boldsymbol{w}_k)|^2\,\|\nabla\bar{\delta}(s, \boldsymbol{\theta}_k)\|_2^2 + 2|\widehat{f}(s, \boldsymbol{w}_k)|^2\,\|\nabla\bar{\delta}(s, \boldsymbol{\theta}_k) - \nabla\widehat{\delta}(s, \boldsymbol{\theta}_k)\|_2^2. \tag{54}$$

Since
$$\nabla \bar{\delta}(s, \boldsymbol{\theta}_k) = \nabla f(s, \boldsymbol{\theta}_k) - \gamma \mathbb{E}_\pi[\nabla f(s', \boldsymbol{\theta}_k)|s], \quad \nabla \widehat{\delta}(s, \boldsymbol{\theta}_k) = \nabla \widehat{f}(s, \boldsymbol{\theta}_k) - \gamma \mathbb{E}_\pi[\nabla \widehat{f}(s', \boldsymbol{\theta}_k)|s], \quad (55)$$

where $\mathbb{E}_\pi[\cdot|s]$ denotes the conditional expectation $\mathbb{E}_{a \sim \pi(\cdot|s), s' \sim \mathrm{P}^a(s,\cdot)}[\cdot]$ using the target policy. With $\|\boldsymbol{x}_s\|_2 = 1$, it can be shown that $\|\nabla \bar{\delta}(s, \boldsymbol{\theta}_k)\|_2 \vee \|\nabla \widehat{\delta}(s, \boldsymbol{\theta}_k)\|_2 \leq (1 + \gamma)$. Moreover,

$$\begin{aligned}
\|\nabla \bar{\delta}(s, \boldsymbol{\theta}_k) - \nabla \widehat{\delta}(s, \boldsymbol{\theta}_k)\|_2^2 &\leq 2\|\nabla f(s, \boldsymbol{\theta}_k) - \nabla \widehat{f}(s, \boldsymbol{\theta}_k)\|_2^2 \\
&\quad + 2\gamma^2 \mathbb{E}_\pi[\|\nabla f(s', \boldsymbol{\theta}_k) - \nabla \widehat{f}(s', \boldsymbol{\theta}_k)\|_2^2|s].
\end{aligned} \quad (56)$$

For any $s \in \mathsf{S}$, we have

$$\nabla f(s, \boldsymbol{\theta}_k) - \nabla \widehat{f}(s, \boldsymbol{\theta}_k) = \begin{pmatrix} (\mathbb{1}\{\langle \boldsymbol{\theta}_k^{(1)}, \boldsymbol{x}_s \rangle > 0\} - \mathbb{1}\{\langle \boldsymbol{\theta}_0^{(1)}, \boldsymbol{x}_s \rangle > 0\}) \, m^{-1/2} b_1 \boldsymbol{x}_s \\ \vdots \\ (\mathbb{1}\{\langle \boldsymbol{\theta}_k^{(m)}, \boldsymbol{x}_s \rangle > 0\} - \mathbb{1}\{\langle \boldsymbol{\theta}_0^{(m)}, \boldsymbol{x}_s \rangle > 0\}) \, m^{-1/2} b_m \boldsymbol{x}_s. \end{pmatrix}. \quad (57)$$

As such,

$$\begin{aligned}
\|\nabla f(s, \boldsymbol{\theta}_k) - \nabla \widehat{f}(s, \boldsymbol{\theta}_k)\|_2^2 &= \frac{1}{m} \sum_{r=1}^m |b_r|^2 |\mathbb{1}\{\langle \boldsymbol{\theta}_k^{(r)}, \boldsymbol{x}_s \rangle > 0\} - \mathbb{1}\{\langle \boldsymbol{\theta}_0^{(r)}, \boldsymbol{x}_s \rangle > 0\}|^2 \|\boldsymbol{x}_s\|_2^2 \\
&\stackrel{(a)}{\leq} \frac{1}{m} \sum_{r=1}^m |\mathbb{1}\{\langle \boldsymbol{\theta}_k^{(r)}, \boldsymbol{x}_s \rangle > 0\} - \mathbb{1}\{\langle \boldsymbol{\theta}_0^{(r)}, \boldsymbol{x}_s \rangle > 0\}|^2 \\
&\stackrel{(b)}{\leq} \frac{1}{m} \sum_{r=1}^m \mathbb{1}\{|\langle \boldsymbol{\theta}_0^{(r)}, \boldsymbol{x}_s \rangle| \leq \|\boldsymbol{\theta}_k^{(r)} - \boldsymbol{\theta}_0^{(r)}\|_2\},
\end{aligned} \quad (58)$$

where (a) uses $|b_r| \leq 1$, $\|\boldsymbol{x}_s\|_2^2 = 1$, and (b) follows from the fact

$$|\mathbb{1}\{\langle \boldsymbol{\theta}_k^{(r)}, \boldsymbol{x}_s \rangle > 0\} - \mathbb{1}\{\langle \boldsymbol{\theta}_0^{(r)}, \boldsymbol{x}_s \rangle > 0\}| \leq \mathbb{1}\{|\langle \boldsymbol{\theta}_0^{(r)}, \boldsymbol{x}_s \rangle| \leq \|\boldsymbol{\theta}_k^{(r)} - \boldsymbol{\theta}_0^{(r)}\|_2\},$$

since $\mathbb{1}\{\langle \boldsymbol{\theta}_k^{(r)}, \boldsymbol{x}_s \rangle > 0\} \neq \mathbb{1}\{\langle \boldsymbol{\theta}_0^{(r)}, \boldsymbol{x}_s \rangle > 0\}$ implies $|\langle \boldsymbol{\theta}_0^{(r)}, \boldsymbol{x}_s \rangle| \leq \|\boldsymbol{\theta}_k^{(r)} - \boldsymbol{\theta}_0^{(r)}\|_2$. Lastly, we observe

$$|\widehat{f}(s, \boldsymbol{w}_k)|^2 \leq 2|\widehat{f}(s, \boldsymbol{\theta}_0)|^2 + 2|\widehat{f}(s, \boldsymbol{w}_k) - \widehat{f}(s, \boldsymbol{\theta}_0)|^2 \leq 2|\widehat{f}(s, \boldsymbol{\theta}_0)|^2 + 2B^2. \quad (59)$$

Using Lemma 4.1, A.3, A.4, we obtain

$$\mathbb{E}_{\mathsf{init}, \mu}[E_k^{(1,1)}] \leq 4(1 + \gamma^2)\{c_0 B^3 + 2c_1 B^3 + 2c_2 B\} m^{-1/2} + 8(1 + \gamma^2) m^{-1} \quad (60)$$

Similarly, we observe

$$E_k^{(1,2)} \leq 2|f(s, \boldsymbol{\theta}_k) - \widehat{f}(s, \boldsymbol{\theta}_k)|^2 \|\nabla f(s, \boldsymbol{\theta}_k)\|_2^2 + 2|\widehat{f}(s, \boldsymbol{\theta}_k)|^2 \|\nabla f(s, \boldsymbol{\theta}_k) - \nabla \widehat{f}(s, \boldsymbol{\theta}_k)\|_2^2. \quad (61)$$

Similar to (59), we have $|\widehat{f}(s, \boldsymbol{\theta}_k)|^2 \leq 2|\widehat{f}(s, \boldsymbol{\theta}_0)|^2 + 2B^2$. Applying similar steps as before yields

$$\mathbb{E}_{\mathsf{init}, \mu}[E_k^{(1,2)}] \leq \big(2c_0 B^3 + 2(2c_1 B^3 + 2c_2 B)\big) m^{-1/2} + 4m^{-1}. \quad (62)$$

Finally, we get

$$\mathbb{E}_{\mathsf{init}}[\|\widehat{e}_k^{(1)}\|_2^2] \leq 2 \mathbb{E}_{\mathsf{init}, \mu}\big[E_k^{(1,1)} + \upsilon^2 E_k^{(1,2)}\big] \leq \widetilde{C}_0 \, m^{-1/2}. \quad (63)$$

The next step is to bound $\mathbb{E}_{\mathsf{init}}[\|\widehat{e}_k^{(2)}\|_2^2]$, we observe that

$$\begin{aligned}
\|\widehat{e}_k^{(2)}\|_2^2 &\leq \mathbb{E}_\mu\Big[\|\big(\bar{\delta}(s, \boldsymbol{\theta}_k) - f(s, \boldsymbol{w}_k)\big)\nabla_{\boldsymbol{w}} f(s, \boldsymbol{w}_k) - \big((\widehat{\delta}(s, \boldsymbol{\theta}_k) - \widehat{f}(s, \boldsymbol{w}_k))\nabla_{\boldsymbol{w}} \widehat{f}(s, \boldsymbol{w}_k)\big)\|_2^2\Big] \\
&\leq 2\mathbb{E}_\mu\Big[|\bar{\delta}(s, \boldsymbol{\theta}_k) - f(s, \boldsymbol{w}_k) - (\widehat{\delta}(s, \boldsymbol{\theta}_k) - \widehat{f}(s, \boldsymbol{w}_k))|^2 \|\nabla f(s, \boldsymbol{w}_k)\|_2^2\Big] \\
&\quad + 2\mathbb{E}_\mu\Big[|\widehat{\delta}(s, \boldsymbol{\theta}_k) - \widehat{f}(s, \boldsymbol{w}_k)|^2 \|\nabla \widehat{f}(s, \boldsymbol{w}_k) - \nabla f(s, \boldsymbol{w}_k)\|_2^2\Big].
\end{aligned} \quad (64)$$

We observe that

$$\mathbb{E}_{\mathsf{init},\mu}\left[|\bar{\delta}(s,\boldsymbol{\theta}_k) - f(s,\boldsymbol{w}_k) - (\widehat{\delta}(s,\boldsymbol{\theta}_k) - \widehat{f}(s,\boldsymbol{w}_k))|^2\|\nabla f(s,\boldsymbol{w}_k)\|_2^2\right]$$

$$\leq \mathbb{E}_{\mathsf{init},\mu}\left[|\bar{\delta}(s,\boldsymbol{\theta}_k) - f(s,\boldsymbol{w}_k) - (\widehat{\delta}(s,\boldsymbol{\theta}_k) - \widehat{f}(s,\boldsymbol{w}_k))|^2\right] \leq 3c_0 B^3 m^{-1/2}, \tag{65}$$

where the last inequality is due to Lemma 4.1 and the fact $\|\nabla f(s,\boldsymbol{w}_k)\|_2^2 \leq 1$. Moreover, since $|\widehat{\delta}(s,\boldsymbol{\theta}_k) - \widehat{f}(s,\boldsymbol{w}_k)|^2 \leq 4(\overline{r}^2 + 3B^2 + 3|\widehat{f}(s,\boldsymbol{\theta}_0)|^2)$, we have

$$\mathbb{E}_{\mathsf{init},\mu}\left[|\widehat{\delta}(s,\boldsymbol{\theta}_k) - \widehat{f}(s,\boldsymbol{w}_k)|^2\|\nabla \widehat{f}(s,\boldsymbol{w}_k) - \nabla f(s,\boldsymbol{w}_k)\|_2^2\right]$$

$$\overset{(a)}{\leq} 4\mathbb{E}_{\mathsf{init},\mu}\left[(\overline{r}^2 + 3B^2 + 3|\widehat{f}(s,\boldsymbol{\theta}_0)|^2)\frac{1}{m}\sum_{r=1}^{m}\mathbb{1}\{|\langle \boldsymbol{\theta}_0^{(r)}, \boldsymbol{x}_s\rangle| \leq \|\boldsymbol{w}_k^{(r)} - \boldsymbol{\theta}_0^{(r)}\|_2\}\right] \tag{66}$$

$$\overset{(b)}{\leq} \left(4c_1\overline{r}^2 B + 12c_1 B^3 + 12c_2 B\right)m^{-1/2} + 12m^{-1},$$

where (a) uses (58) and (b) is due to Lemma A.3. Combining the above inequalities also yields

$$\mathbb{E}_{\mathsf{init}}[\|\widehat{\boldsymbol{e}}_k^{(2)}\|_2^2] \leq \widetilde{\mathrm{C}}_0\, m^{-1/2}. \tag{67}$$

The proof is concluded.

## A.2 Proof of Lemma A.2

We should use the shorthand notation $\mathbb{E}_\mu[\cdot]$ to denote that the expectation is taken w.r.t. $s \sim \mu^\Pi$, as well as $a \sim \Pi(\cdot|s), s' \sim \mathrm{P}^a(s,\cdot)$. Define the following mean field matrix/vectors:

$$\boldsymbol{A}_0 = \mathbb{E}_\mu\big[\ell(\boldsymbol{x}_s)\ell(\boldsymbol{x}_s)^\top\big], \quad \boldsymbol{A}_1 = \mathbb{E}_\mu\big[\rho(a|s)\ell(\boldsymbol{x}_s)\ell(\boldsymbol{x}_{s'})^\top\big], \quad \boldsymbol{b} = \mathbb{E}_\mu\big[\rho(a|s)\mathrm{R}(s,a)\ell(\boldsymbol{x}_s)\big],$$

and we further define $\widetilde{\boldsymbol{A}}_1 := \boldsymbol{A}_0 - \gamma \boldsymbol{A}_1$. The objective function can then be written as:

$$\widehat{J}_\upsilon(\boldsymbol{\theta}, \boldsymbol{w}) = \boldsymbol{w}^\top \widetilde{\boldsymbol{A}}_1 \boldsymbol{\theta} - \boldsymbol{w}^\top \boldsymbol{b} - \frac{1}{2}\boldsymbol{w}^\top \boldsymbol{A}_0 \boldsymbol{w} + \frac{\upsilon}{2}\boldsymbol{\theta}^\top \boldsymbol{A}_0 \boldsymbol{\theta}.$$

A direct computation of the gradient of the above leads to

$$\langle \widehat{\Phi}(\boldsymbol{z}_k) - \widehat{\Phi}(\widehat{\boldsymbol{z}}), \boldsymbol{z}_k - \widehat{\boldsymbol{z}}\rangle = \upsilon(\boldsymbol{\theta}_k - \widehat{\boldsymbol{\theta}})^\top \boldsymbol{A}_0(\boldsymbol{\theta}_k - \widehat{\boldsymbol{\theta}}) + (\boldsymbol{w}_k - \widehat{\boldsymbol{w}})^\top \boldsymbol{A}_0(\boldsymbol{w}_k - \widehat{\boldsymbol{w}})$$

$$= \upsilon\,\mathbb{E}_\mu\big[(\boldsymbol{\theta}_k - \widehat{\boldsymbol{\theta}})^\top \ell(s)\ell(s)^\top(\boldsymbol{\theta}_k - \widehat{\boldsymbol{\theta}})\big] + \mathbb{E}_\mu\big[(\boldsymbol{w}_k - \widehat{\boldsymbol{w}})^\top \ell(s)\ell(s)^\top(\boldsymbol{w}_k - \widehat{\boldsymbol{w}})\big]$$

$$= \upsilon\,\mathbb{E}_\mu\big[|\widehat{f}(s,\boldsymbol{\theta}_k) - \widehat{f}(s,\widehat{\boldsymbol{\theta}})|^2\big] + \mathbb{E}_\mu\big[|\widehat{f}(s,\boldsymbol{w}_k) - \widehat{f}(s,\widehat{\boldsymbol{w}})|^2\big] \geq \min\{1, \upsilon\}\, d_{\widehat{f}}(\boldsymbol{z}_k, \widehat{\boldsymbol{z}}),$$

where the last equality is due to the definition of $\widehat{f}(\cdot, \cdot)$ in (14). This proves the first inequality of the lemma.

For the second identity, we observe that

$$\begin{pmatrix} \nabla_{\boldsymbol{\theta}}\widehat{J}_\upsilon(\boldsymbol{z}_k) - \nabla_{\boldsymbol{\theta}}\widehat{J}_\upsilon(\widehat{\boldsymbol{z}}) \\ \nabla_{\boldsymbol{w}}\widehat{J}_\upsilon(\boldsymbol{z}_k) - \nabla_{\boldsymbol{w}}\widehat{J}_\upsilon(\widehat{\boldsymbol{z}}) \end{pmatrix} = \begin{pmatrix} \widetilde{\boldsymbol{A}}_1^\top(\boldsymbol{w}_k - \widehat{\boldsymbol{w}}) + \upsilon\,\boldsymbol{A}_0(\boldsymbol{\theta}_k - \widehat{\boldsymbol{\theta}}) \\ \widetilde{\boldsymbol{A}}_1(\boldsymbol{\theta}_k - \widehat{\boldsymbol{\theta}}) - \boldsymbol{A}_0(\boldsymbol{w}_k - \widehat{\boldsymbol{w}}) \end{pmatrix}.$$

This yields

$$\|\nabla \widehat{J}_\upsilon(\boldsymbol{z}_k) - \nabla \widehat{J}_\upsilon(\widehat{\boldsymbol{z}})\|_2 \leq \|\widetilde{\boldsymbol{A}}_1^\top(\boldsymbol{w}_k - \widehat{\boldsymbol{w}})\|_2 + \upsilon\,\|\boldsymbol{A}_0(\boldsymbol{\theta}_k - \widehat{\boldsymbol{\theta}})\|_2 + \|\widetilde{\boldsymbol{A}}_1(\boldsymbol{\theta}_k - \widehat{\boldsymbol{\theta}})\|_2 + \|\boldsymbol{A}_0(\boldsymbol{w}_k - \widehat{\boldsymbol{w}})\|_2.$$

We have

$$\|\boldsymbol{A}_0(\boldsymbol{\theta}_k - \widehat{\boldsymbol{\theta}})\|_2 = \big\|\mathbb{E}_\mu\big[\ell(\boldsymbol{x}_s)(\widehat{f}(s,\boldsymbol{\theta}_k) - \widehat{f}(s,\widehat{\boldsymbol{\theta}}))\big]\big\|_2 \leq \mathbb{E}_\mu\big[\|\ell(\boldsymbol{x}_s)(\widehat{f}(s,\boldsymbol{\theta}_k) - \widehat{f}(s,\widehat{\boldsymbol{\theta}}))\|_2\big]$$

$$\leq \mathbb{E}_\mu\big[|\widehat{f}(s,\boldsymbol{\theta}_k) - \widehat{f}(s,\widehat{\boldsymbol{\theta}})|\,\|\ell(\boldsymbol{x}_s)\|_2\big] \leq \sqrt{\mathbb{E}_\mu\big[|\widehat{f}(s,\boldsymbol{\theta}_k) - \widehat{f}(s,\widehat{\boldsymbol{\theta}})|^2\big]}\sqrt{\mathbb{E}_\mu[\|\ell(\boldsymbol{x}_s)\|_2^2]},$$

where we have applied the Cauchy-Schwarz inequality in the last step. Also,

$$\|\widetilde{\boldsymbol{A}}_1^\top(\boldsymbol{w}_k - \widehat{\boldsymbol{w}})\|_2 = \|(\boldsymbol{A}_0 - \gamma \boldsymbol{A}_1^\top)(\boldsymbol{w}_k - \widehat{\boldsymbol{w}})\|_2$$

$$= \|\mathbb{E}_\mu[(\ell(\boldsymbol{x}_s) - \gamma\rho(a|s)\ell(\boldsymbol{x}_{s'}))(\widehat{f}(s,\boldsymbol{w}_k) - \widehat{f}(s,\widehat{\boldsymbol{w}}))]\|_2$$

$$\overset{(a)}{\leq} \mathbb{E}_{s\sim\mu^\Pi, a\sim\pi(\cdot|s), s'\sim\mathrm{P}^a(\cdot|s)}\big[|\widehat{f}(s,\boldsymbol{w}_k) - \widehat{f}(s,\widehat{\boldsymbol{w}})|\,\|\ell(\boldsymbol{x}_s) - \gamma\ell(\boldsymbol{x}_{s'})\|_2\big]$$

$$\leq \mathbb{E}_\mu\big[|\widehat{f}(s,\boldsymbol{w}_k) - \widehat{f}(s,\widehat{\boldsymbol{w}})|\,\|\ell(\boldsymbol{x}_s)\|_2\big] + \gamma\mathbb{E}_{s\sim\mu^\Pi, a\sim\pi(\cdot|s), s'\sim\mathrm{P}^a(\cdot|s)}\big[|\widehat{f}(s,\boldsymbol{w}_k) - \widehat{f}(s,\widehat{\boldsymbol{w}})|\,\|\ell(\boldsymbol{x}_{s'})\|_2\big]$$

$$\overset{(b)}{\leq} (1+\gamma)\sqrt{\mathbb{E}_\mu\big[|\widehat{f}(s,\boldsymbol{w}_k) - \widehat{f}(s,\widehat{\boldsymbol{w}})|^2\big]},$$

where the expectation in (a) is switched to the target policy through evaluating the expectation with the importance ratio, and (b) uses $\|\ell(\boldsymbol{x}_s)\|_2^2 \le 1$.

Similarly, we get

$$
\begin{aligned}
\|\widetilde{\boldsymbol{A}}_1(\boldsymbol{\theta}_k - \widehat{\boldsymbol{\theta}})\|_2 &= \|(\boldsymbol{A}_0 - \gamma\boldsymbol{A}_1)(\boldsymbol{\theta}_k - \widehat{\boldsymbol{\theta}})\|_2 \le \|\boldsymbol{A}_0(\boldsymbol{\theta}_k - \widehat{\boldsymbol{\theta}})\|_2 + \gamma\|\boldsymbol{A}_1(\boldsymbol{\theta}_k - \widehat{\boldsymbol{\theta}})\|_2 \\
&= \left\|\mathbb{E}_\mu\big[\ell(\boldsymbol{x}_s)(\widehat{f}(s,\boldsymbol{\theta}_k) - \widehat{f}(s,\widehat{\boldsymbol{\theta}}))\big]\right\|_2 + \gamma\left\|\mathbb{E}_{s\sim\mu^\Pi, a\sim\pi(\cdot|s), s'\sim\mathrm{P}^a(\cdot|s)}\big[\ell(\boldsymbol{x}_s)(\widehat{f}(s',\boldsymbol{\theta}_k) - \widehat{f}(s',\widehat{\boldsymbol{\theta}}))\big]\right\|_2 \\
&\le \mathbb{E}_\mu\big[|\widehat{f}(s,\boldsymbol{\theta}_k) - \widehat{f}(s,\widehat{\boldsymbol{\theta}})|\|\ell(\boldsymbol{x}_s)\|_2\big] + \gamma\,\mathbb{E}_{s\sim\mu^\Pi, a\sim\pi(\cdot|s), s'\sim\mathrm{P}^a(\cdot|s)}\big[|\widehat{f}(s',\boldsymbol{\theta}_k) - \widehat{f}(s',\widehat{\boldsymbol{\theta}})|\|\ell(\boldsymbol{x}_s)\|_2\big] \\
&\le (1+\gamma)\sqrt{\big[|\widehat{f}(s,\boldsymbol{\theta}_k) - \widehat{f}(s,\widehat{\boldsymbol{\theta}})|^2\big]}.
\end{aligned}
$$

Therefore,

$$
\begin{aligned}
&\|\nabla\widehat{J}_\upsilon(\boldsymbol{z}_k) - \nabla\widehat{J}_\upsilon(\widehat{\boldsymbol{z}})\|_2^2 \\
&\le 4\Big((1 + (1+\gamma)^2)\mathbb{E}_\mu\big[|\widehat{f}(s,\boldsymbol{w}_k) - \widehat{f}(s,\widehat{\boldsymbol{w}})|^2\big] + (\upsilon^2 + (1+\gamma)^2)\mathbb{E}_\mu\big[|\widehat{f}(s,\boldsymbol{\theta}_k) - \widehat{f}(s,\widehat{\boldsymbol{\theta}})|^2\big]\Big) \\
&\le 4\{(1\vee\upsilon)^2 + (1+\gamma)^2\}\,d_{\widehat{f}}(\boldsymbol{z}_k,\widehat{\boldsymbol{z}}).
\end{aligned}
$$

This concludes the proof.

### A.3  Proof of Lemma A.4

Observe the following expansion

$$
\begin{aligned}
|\widehat{f}(s,\boldsymbol{\theta}_0)|^2 &= \frac{1}{m}\Big(\sum_{r=1}^m b_r\mathbb{1}\{\langle\boldsymbol{\theta}_0^{(r)}, \boldsymbol{x}_s\rangle > 0\}\langle\boldsymbol{\theta}_0^{(r)}, \boldsymbol{x}_s\rangle\Big)^2 \\
&\le \frac{1}{m}\left(\sum_{r=1}^m \mathbb{1}\{\langle\boldsymbol{\theta}_0^{(r)}, \boldsymbol{x}_s\rangle > 0\}|\langle\boldsymbol{\theta}_0^{(r)}, \boldsymbol{x}_s\rangle|^2 + \sum_{r=1}^m\sum_{r'\ne r} b_r b_{r'}\mathbb{1}\{\langle\boldsymbol{\theta}_0^{(r)}, \boldsymbol{x}_s\rangle > 0\}\langle\boldsymbol{\theta}_0^{(r)}, \boldsymbol{x}_s\rangle\mathbb{1}\{\langle\boldsymbol{\theta}_0^{(r')}, \boldsymbol{x}_s\rangle > 0\}\langle\boldsymbol{\theta}_0^{(r')}, \boldsymbol{x}_s\rangle\right) \\
&\le \frac{1}{m}\left(\sum_{r=1}^m \|\boldsymbol{\theta}_0^{(r)}\|_2^2 + \sum_{r=1}^m\sum_{r'\ne r} b_r b_{r'}\mathbb{1}\{\langle\boldsymbol{\theta}_0^{(r)}, \boldsymbol{x}_s\rangle > 0\}\langle\boldsymbol{\theta}_0^{(r)}, \boldsymbol{x}_s\rangle\mathbb{1}\{\langle\boldsymbol{\theta}_0^{(r')}, \boldsymbol{x}_s\rangle > 0\}\langle\boldsymbol{\theta}_0^{(r')}, \boldsymbol{x}_s\rangle\right).
\end{aligned}
$$

Together with the above expansion and using the fact that $\mathbb{E}_{\mathsf{init}}[b_r b_{r'}] = 0$ for any $r \ne r'$, the desired expectation with respect to the NN initialization can be computed as

$$
\begin{aligned}
&\mathbb{E}_{\mathsf{init},\mu}\left[|\widehat{f}(s,\boldsymbol{\theta}_0)|^2\frac{1}{m}\sum_{r=1}^m \mathbb{1}\{|\langle\boldsymbol{\theta}_0^{(r)}, \boldsymbol{x}_s\rangle| \le \|\boldsymbol{\theta}^{(r)} - \boldsymbol{\theta}_0^{(r)}\|_2\}\right] \\
&\le \frac{1}{m^2}\mathbb{E}_{\mathsf{init},\mu}\left[\left(\sum_{r=1}^m \|\boldsymbol{\theta}_0^{(r)}\|_2^2\right)\left(\sum_{r=1}^m \mathbb{1}\{|\langle\boldsymbol{\theta}_0^{(r)}, \boldsymbol{x}_s\rangle| \le \|\boldsymbol{\theta}^{(r)} - \boldsymbol{\theta}_0^{(r)}\|_2\}\right)\right] \\
&\le \frac{1}{m^2}\mathbb{E}_{\mathsf{init},\mu}\left[\sum_{r=1}^m \|\boldsymbol{\theta}_0^{(r)}\|_2^2\sum_{r'\ne r}\mathbb{1}\{|\langle\boldsymbol{\theta}_0^{(r')}, \boldsymbol{x}_s\rangle| \le \|\boldsymbol{\theta}^{(r')} - \boldsymbol{\theta}_0^{(r')}\|_2\} + \sum_{r=1}^m \|\boldsymbol{\theta}_0^{(r)}\|_2^2\mathbb{1}\{|\langle\boldsymbol{\theta}_0^{(r)}, \boldsymbol{x}_s\rangle| \le \|\boldsymbol{\theta}^{(r)} - \boldsymbol{\theta}_0^{(r)}\|_2\}\right].
\end{aligned}
$$

Let us study the first term inside the expectation above:

$$
\begin{aligned}
&\frac{1}{m^2}\mathbb{E}_{\mathsf{init},\mu}\left[\sum_{r=1}^m \|\boldsymbol{\theta}_0^{(r)}\|_2^2\sum_{r'\ne r}\mathbb{1}\{|\langle\boldsymbol{\theta}_0^{(r')}, \boldsymbol{x}_s\rangle| \le \|\boldsymbol{\theta}^{(r')} - \boldsymbol{\theta}_0^{(r')}\|_2\}\right] \\
&\overset{(a)}{=} \frac{1}{m}\sum_{r=1}^m \mathbb{E}_{\mathsf{init},\mu}[\|\boldsymbol{\theta}_0^{(r)}\|_2^2]\,\mathbb{E}_{\mathsf{init},\mu}\left[\frac{1}{m}\sum_{r'\ne r}\mathbb{1}\{|\langle\boldsymbol{\theta}_0^{(r')}, \boldsymbol{x}_s\rangle| \le \|\boldsymbol{\theta}^{(r')} - \boldsymbol{\theta}_0^{(r')}\|_2\}\right] \\
&\le \frac{1}{m}\sum_{r=1}^m \mathbb{E}_{\mathsf{init},\mu}[\|\boldsymbol{\theta}_0^{(r)}\|_2^2]\,\mathbb{E}_{\mathsf{init},\mu}\left[\frac{1}{m}\sum_{r'=1}^m\mathbb{1}\{|\langle\boldsymbol{\theta}_0^{(r')}, \boldsymbol{x}_s\rangle| \le \|\boldsymbol{\theta}^{(r')} - \boldsymbol{\theta}_0^{(r')}\|_2\}\right] \overset{(b)}{\le} c_1 B m^{-1/2},
\end{aligned}
$$

$$(68)$$

where (a) is due to the independence between $\boldsymbol{\theta}_0^{(r)}$ and $\boldsymbol{\theta}_0^{(r')}$ with $r' \neq r$ and (b) is due to Lemma A.3. For the second term, we observe

$$\frac{1}{m^2}\mathbb{E}_{\text{init},\mu}\left[\sum_{r=1}^{m}\|\boldsymbol{\theta}_0^{(r)}\|_2^2\mathbb{1}\{|\langle\boldsymbol{\theta}_0^{(r)},\boldsymbol{x}_s\rangle|\leq\|\boldsymbol{\theta}^{(r)}-\boldsymbol{\theta}_0^{(r)}\|_2\}\right]\leq\frac{1}{m^2}\sum_{r=1}^{m}\mathbb{E}_{\text{init}}[\|\boldsymbol{\theta}_0^{(r)}\|_2^2]=\frac{1}{m}. \quad (69)$$

Combining (68) and (69) leads to the desired result.

## B  Proof of Theorem 3.2

Note that the stochastic neural GTD algorithm is similar to the population GTD algorithm except for the additional error term $\boldsymbol{e}_k^{(i)}$ in the updates [cf. (22)]. In particular, with respect to (15), the stochastic neural GTD algorithm in Algorithm 1 can be written as:

$$\begin{aligned}
\boldsymbol{\theta}_{k+1} &= \mathcal{P}_{S_B}\left\{\boldsymbol{\theta}_k - \beta_k\left(\nabla_{\boldsymbol{\theta}}\widehat{J}_\upsilon(\boldsymbol{z}_k) + \widehat{\boldsymbol{e}}_k^{(1)} + \boldsymbol{e}_k^{(1)}\right)\right\}, \\
\boldsymbol{w}_{k+1} &= \mathcal{P}_{S_B}\left\{\boldsymbol{w}_k + \beta_k\left(\nabla_{\boldsymbol{w}}\widehat{J}_\upsilon(\boldsymbol{z}_k) + \widehat{\boldsymbol{e}}_k^{(2)} + \boldsymbol{e}_k^{(2)}\right)\right\},
\end{aligned} \quad (70)$$

We proceed by observing that (42), (43) can be modified as

$$\begin{aligned}
\|\boldsymbol{\theta}_{k+1} - \widehat{\boldsymbol{\theta}}\|_2^2 &\leq \|\boldsymbol{\theta}_k - \widehat{\boldsymbol{\theta}}\|_2^2 + \beta_k^2\|\nabla_{\boldsymbol{\theta}}\widehat{J}_\upsilon(\boldsymbol{z}_k) + \widehat{\boldsymbol{e}}_k^{(1)} + \boldsymbol{e}_k^{(1)} - \nabla_{\boldsymbol{\theta}}\widehat{J}_\upsilon(\widehat{\boldsymbol{z}})\|_2^2 \\
&\quad - 2\beta_k\langle\boldsymbol{\theta}_k - \widehat{\boldsymbol{\theta}}, \nabla_{\boldsymbol{\theta}}\widehat{J}_\upsilon(\boldsymbol{z}_k) + \widehat{\boldsymbol{e}}_k^{(1)} + \boldsymbol{e}_k^{(1)} - \nabla_{\boldsymbol{\theta}}\widehat{J}_\upsilon(\widehat{\boldsymbol{z}})\rangle,
\end{aligned}$$

$$\begin{aligned}
\|\boldsymbol{w}_{k+1} - \widehat{\boldsymbol{w}}\|_2^2 &\leq \|\boldsymbol{w}_k - \widehat{\boldsymbol{w}}\|_2^2 + \beta_k^2\|\nabla_{\boldsymbol{w}}\widehat{J}_\upsilon(\boldsymbol{z}_k) + \widehat{\boldsymbol{e}}_k^{(2)} + \boldsymbol{e}_k^{(2)} - \nabla_{\boldsymbol{w}}\widehat{J}_\upsilon(\widehat{\boldsymbol{z}})\|_2^2 \\
&\quad + 2\beta_k\langle\boldsymbol{w}_k - \widehat{\boldsymbol{w}}, \nabla_{\boldsymbol{w}}\widehat{J}_\upsilon(\boldsymbol{z}_k) + \widehat{\boldsymbol{e}}_k^{(2)} + \boldsymbol{e}_k^{(2)} - \nabla_{\boldsymbol{w}}\widehat{J}_\upsilon(\widehat{\boldsymbol{z}})\rangle.
\end{aligned}$$

Note that when conditioned on $\mathcal{F}_{k-1} = \sigma\{\boldsymbol{\theta}_0, \widetilde{s}_0, ..., \widetilde{s}_{k-1}\}$, the iterate $\boldsymbol{z}_k = (\boldsymbol{\theta}_k, \boldsymbol{w}_k)$ is deterministic. As such, using H3 that the noise $\boldsymbol{e}_k^{(i)}$ is zero-mean when conditioned on $\mathcal{F}_{k-1}$, $\mathbb{E}[\langle\boldsymbol{\theta}_k - \widehat{\boldsymbol{\theta}}, \boldsymbol{e}_k^{(1)}\rangle \,|\, \mathcal{F}_{k-1}] = \mathbb{E}[\langle\boldsymbol{w}_k - \widehat{\boldsymbol{w}}, \boldsymbol{e}_k^{(2)}\rangle \,|\, \mathcal{F}_{k-1}] = 0$.

It follows that for any $c_1 > 0$, we have

$$\begin{aligned}
\mathbb{E}[\|\boldsymbol{z}_{k+1} - \widehat{\boldsymbol{z}}\|_2^2|\mathcal{F}_{k-1}] &\leq \|\boldsymbol{z}_k - \widehat{\boldsymbol{z}}\|_2^2 - 2\beta_k\langle\widehat{\Phi}(\boldsymbol{z}_k) - \widehat{\Phi}(\widehat{\boldsymbol{z}}), \boldsymbol{z}_k - \widehat{\boldsymbol{z}}\rangle + \frac{\beta_k}{c_1}\|\boldsymbol{z}_k - \widehat{\boldsymbol{z}}\|_2^2 \\
&\quad + 3\beta_k^2\|\nabla\widehat{J}_\upsilon(\boldsymbol{z}_k) - \nabla\widehat{J}_\upsilon(\widehat{\boldsymbol{z}})\|_2^2 + (3\beta_k^2 + c_1\beta_k)\{\|\widehat{\boldsymbol{e}}_k^{(1)}\|_2^2 + \|\widehat{\boldsymbol{e}}_k^{(2)}\|_2^2\} \\
&\quad + 3\beta_k^2\,\mathbb{E}[\|\boldsymbol{e}_k^{(1)}\|_2^2 + \|\boldsymbol{e}_k^{(2)}\|_2^2|\mathcal{F}_{k-1}] \\
&\leq \|\boldsymbol{z}_k - \widehat{\boldsymbol{z}}\|_2^2 - 2\beta_k\mu d_{\widehat{f}}(\boldsymbol{z}_k, \widehat{\boldsymbol{z}}) + \frac{\beta_k}{c_1}4B^2 + 3\beta_k^2 L_\Phi d_{\widehat{f}}(\boldsymbol{z}_k, \widehat{\boldsymbol{z}}) \\
&\quad + (3\beta_k^2 + c_1\beta_k)\{\|\widehat{\boldsymbol{e}}_k^{(1)}\|_2^2 + \|\widehat{\boldsymbol{e}}_k^{(2)}\|_2^2\} + 6\beta_k^2\sigma^2,
\end{aligned} \quad (71)$$

where the last inequality used $\|\boldsymbol{z}_k - \widehat{\boldsymbol{z}}\|_2^2 \leq 4B^2$ and Lemma A.2. Rearranging terms and using our conditions on the step size such that $\mu - \frac{3}{2}\beta_k L_\Phi \geq \frac{\mu}{2}$, these lead to

$$\begin{aligned}
\beta_k\mu\, d_{\widehat{f}}(\boldsymbol{z}_k, \widehat{\boldsymbol{z}}) &\leq \|\boldsymbol{z}_k - \widehat{\boldsymbol{z}}\|_2^2 - \mathbb{E}[\|\boldsymbol{z}_{k+1} - \widehat{\boldsymbol{z}}\|_2^2|\mathcal{F}_{k-1}] + 4B^2\left(\frac{\beta_k}{c_1}\right) \\
&\quad + (3\beta_k^2 + c_1\beta_k)\{\|\widehat{\boldsymbol{e}}_k^{(1)}\|_2^2 + \|\widehat{\boldsymbol{e}}_k^{(2)}\|_2^2\} + 6\beta_k^2\sigma^2\}.
\end{aligned} \quad (72)$$

We take $c_1 = m^{\frac{1}{4}}$ and invoke Lemma A.1, this lead to

$$\beta_k\mu\, d_{\widehat{f}}(\boldsymbol{z}_k, \widehat{\boldsymbol{z}}) \leq \|\boldsymbol{z}_k - \widehat{\boldsymbol{z}}\|_2^2 - \mathbb{E}[\|\boldsymbol{z}_{k+1} - \widehat{\boldsymbol{z}}\|_2^2|\mathcal{F}_{k-1}] + \frac{\beta_k}{m^{\frac{1}{4}}}\left(2\mathrm{C}_0 B^3 + 4B^2\right) + 2\beta_k^2\left\{\frac{3\mathrm{C}_0 B^3}{\sqrt{m}} + 3\sigma^2\right\}. \quad (73)$$

Taking total expectations on both sides and summing up the inequality from $k = 0$ to $k = n$ leads to

$$\mu\sum_{k=0}^{n}\beta_k\mathbb{E}[d_f(\boldsymbol{z}_k, \widehat{\boldsymbol{z}})] \leq \mathbb{E}[\|\boldsymbol{z}_0 - \widehat{\boldsymbol{z}}\|_2^2] + \frac{2\mathrm{C}_0 B^3 + 4B^2}{m^{\frac{1}{4}}}\sum_{k=0}^{n}\beta_k + 2\left\{\frac{3\mathrm{C}_0 B^3}{\sqrt{m}} + 3\sigma^2\right\}\sum_{k=0}^{n}\beta_k^2. \quad (74)$$

Recall that $I_n \in \{0, ..., n\}$ is an independent random variable chosen from the distribution $\mathbb{P}(I_n = k) = \beta_k / \sum_{\ell=0}^{n} \beta_\ell$, we have

$$\mathbb{E}\big[d_f(\boldsymbol{z}_{I_n}, \widehat{\boldsymbol{z}})\big] \leq \frac{2\mathrm{C}_0 B^3 + 4B^2}{\mu m^{\frac{1}{4}}} + \frac{\mathbb{E}[\|\boldsymbol{z}_0 - \widehat{\boldsymbol{z}}\|_2^2] + 2\big\{\frac{3\mathrm{C}_0 B^3}{\sqrt{m}} + 3\sigma^2\big\} \sum_{k=0}^{n} \beta_k^2}{\mu \sum_{k=0}^{n} \beta_k}. \tag{75}$$

This concludes the proof of our theorem.

## C    Proof of Theorem 3.4

Note that $\mathbb{E}_\mu[\cdot] = \mathbb{E}_{s \sim \mu^\Pi}[\cdot]$. We begin the proof by recalling that, for any $\boldsymbol{\theta} \in S_B$,

$$J_\upsilon(\boldsymbol{\theta}) = \frac{1}{2}\mathbb{E}_\mu\big[\bar{\delta}(s, \boldsymbol{\theta})^2 + \upsilon f(s, \boldsymbol{\theta})^2\big], \tag{76}$$

where we define that (the importance ratio $\rho(a|s)$ has been absorbed in the conditional expectation)

$$\bar{\delta}(s, \boldsymbol{\theta}) = f(s, \boldsymbol{\theta}) - \mathrm{r}^\pi(s) - \gamma \mathrm{P}^\pi f(s, \boldsymbol{\theta}),$$

and we have defined $\mathrm{r}^\pi(s) = \mathbb{E}_{a \sim \pi(\cdot|s)}[\mathrm{R}(s, a)]$. Moreover, we define the following using the linearized NN function:

$$\widehat{\delta}(s, \boldsymbol{\theta}) = \widehat{f}(s, \boldsymbol{\theta}) - \mathrm{r}^\pi(s) - \gamma \mathrm{P}^\pi \widehat{f}(s, \boldsymbol{\theta}).$$

**Estimates on the differences between $\bar{\delta}, \widehat{\delta}, f, \widehat{f}$**    We first derive a few upper bounds as follows. Observe from Lemma 4.1 that for any $\boldsymbol{\theta} \in S_B$,

$$\mathbb{E}_{\mathsf{init}, \mu}[|\bar{\delta}(s, \boldsymbol{\theta}) - \widehat{\delta}(s, \boldsymbol{\theta})|^2] \leq 2(1 + \bar{\rho}^2 \gamma^2) c_0 B^3 m^{-1/2} =: R_\delta^{(0)}$$

as well as $\mathbb{E}_{\mathsf{init}, \mu}[|f(s, \boldsymbol{\theta}) - \widehat{f}(s, \boldsymbol{\theta})|^2] \leq c_0 B^3 m^{-1/2} =: R_f^{(0)}$. Now, consider that

$$|f(s, \boldsymbol{\theta})|^2 \leq \underbrace{|f(s, \boldsymbol{\theta}) - \widehat{f}(s, \boldsymbol{\theta})|^2 + 2|f(s, \boldsymbol{\theta}) - \widehat{f}(s, \boldsymbol{\theta})||\widehat{f}(s, \boldsymbol{\theta})|}_{=: \bar{R}_f^{(1)}(s, \boldsymbol{\theta})} + |\widehat{f}(s, \boldsymbol{\theta})|^2,$$

$$|\bar{\delta}(s, \boldsymbol{\theta})|^2 \leq \underbrace{|\bar{\delta}(s, \boldsymbol{\theta}) - \widehat{\delta}(s, \boldsymbol{\theta})|^2 + 2|\bar{\delta}(s, \boldsymbol{\theta}) - \widehat{\delta}(s, \boldsymbol{\theta})||\widehat{\delta}(s, \boldsymbol{\theta})|}_{=: \bar{R}_\delta^{(1)}(s, \boldsymbol{\theta})} + |\widehat{\delta}(s, \boldsymbol{\theta})|^2,$$

$$|f(s, \boldsymbol{\theta})|^2 \leq \underbrace{|f(s, \boldsymbol{\theta}) - f(s, \widehat{\boldsymbol{\theta}})|^2 + 2|f(s, \boldsymbol{\theta}) - f(s, \widehat{\boldsymbol{\theta}})||f(s, \widehat{\boldsymbol{\theta}})|}_{=: \bar{R}_f^{(2)}(s, \boldsymbol{\theta}, \widehat{\boldsymbol{\theta}})} + |f(s, \widehat{\boldsymbol{\theta}})|^2$$

$$|\delta(s, \boldsymbol{\theta})|^2 \leq \underbrace{|\delta(s, \boldsymbol{\theta}) - \delta(s, \widehat{\boldsymbol{\theta}})|^2 + 2|\delta(s, \boldsymbol{\theta}) - \delta(s, \widehat{\boldsymbol{\theta}})||\delta(s, \widehat{\boldsymbol{\theta}})|}_{=: \bar{R}_\delta^{(2)}(s, \boldsymbol{\theta}, \widehat{\boldsymbol{\theta}})} + |\delta(s, \widehat{\boldsymbol{\theta}})|^2,$$

Note that $\bar{R}_\delta^{(1)}, \bar{R}_f^{(1)}, \bar{R}_\delta^{(2)}, \bar{R}_f^{(2)}$ are measurable functions on $\mathsf{S}$ and depend on $\boldsymbol{\theta}, \widehat{\boldsymbol{\theta}}$. We upper bound these terms as follows.

For $\bar{R}_f^{(1)}$, due to the Cauch-Schwarz inequality, for any $\boldsymbol{\theta} \in S_B$, we have

$$\mathbb{E}_{\mathsf{init}, \mu}\big[|f(s, \boldsymbol{\theta}) - \widehat{f}(s, \boldsymbol{\theta})| \, |\widehat{f}(s, \boldsymbol{\theta})|\big] \leq \Big(\mathbb{E}_{\mathsf{init}, \mu}\big[|\widehat{f}(s, \boldsymbol{\theta})|^2\big]\Big)^{\frac{1}{2}} \Big(\mathbb{E}_{\mathsf{init}, \mu}\big[|f(s, \boldsymbol{\theta}) - \widehat{f}(s, \boldsymbol{\theta})|^2\big]\Big)^{\frac{1}{2}}.$$

As we have $|\widehat{f}(s, \boldsymbol{\theta})|^2 \leq 2|\widehat{f}(s, \boldsymbol{\theta}_0)|^2 + 2B^2$, and

$$\mathbb{E}_{\mathsf{init}, \mu}\big[|\widehat{f}(s, \boldsymbol{\theta}_0)|^2\big] = \mathbb{E}_{\mathsf{init}, \mu}\left[\frac{1}{m}\sum_{r=1}^{m}\sum_{r'=1}^{m} b_r b_{r'} \mathbb{1}\{\langle \boldsymbol{\theta}_0^{(r)}, \boldsymbol{x}_s \rangle > 0\} \mathbb{1}\{\langle \boldsymbol{\theta}_0^{(r')}, \boldsymbol{x}_s \rangle > 0\} \langle \boldsymbol{\theta}_0^{(r)}, \boldsymbol{x}_s \rangle \langle \boldsymbol{\theta}_0^{(r')}, \boldsymbol{x}_s \rangle\right]$$

$$\leq \frac{1}{m}\mathbb{E}_{\mathsf{init}, \mu}\left[\sum_{r=1}^{m} \mathbb{1}\{\langle \boldsymbol{\theta}_0^{(r)}, \boldsymbol{x}_s \rangle > 0\} \|\boldsymbol{\theta}_0^{(r)}\|_2^2\right] \leq 1,$$

where we have used that $b_r$ is independent from $b_{r'}$ if $r \neq r'$ in the first inequality, also see Appendix A.3. The above implies $\mathbb{E}_{\text{init},\mu}\big[|\widehat{f}(s,\boldsymbol{\theta})|^2\big] \leq 2(1+B^2)$, and thus showing that

$$\mathbb{E}_{\text{init},\mu}[\bar{R}_f^{(1)}] \leq c_0 B^3 m^{-1/2} + \sqrt{8c_0(1+B^2)B^3}\, m^{-1/4} =: R_f^{(1)}. \tag{77}$$

For $\bar{R}_\delta^{(1)}$, for any $\boldsymbol{\theta} \in S_B$, applying the Cauch-Schwarz inequality yields

$$\mathbb{E}_{\text{init},\mu}\big[|\delta(s,\boldsymbol{\theta}) - \widehat{\delta}(s,\boldsymbol{\theta})|\,|\widehat{\delta}(s,\boldsymbol{\theta})|\big] \leq \Big(\mathbb{E}_{\text{init},\mu}\big[|\widehat{\delta}(s,\boldsymbol{\theta})|^2\big]\Big)^{\frac{1}{2}} \Big(\mathbb{E}_{\text{init},\mu}\big[|\delta(s,\boldsymbol{\theta}) - \widehat{\delta}(s,\boldsymbol{\theta})|^2\big]\Big)^{\frac{1}{2}}. \tag{78}$$

The above implies

$$\begin{aligned}
\mathbb{E}_{\text{init},\mu}\big[|\widehat{\delta}(s,\boldsymbol{\theta})|^2\big] &\leq 3\bar{r}^2 + 3\,\mathbb{E}_{\text{init},\mu}[|\widehat{f}(s,\boldsymbol{\theta})|^2] + 3\gamma^2\,\mathbb{E}_{\text{init},\mu^\pi}[|\widehat{f}(s',\boldsymbol{\theta})|^2] \\
&\leq 3(\bar{r}^2 + 2(1+\bar{\rho}^2\gamma^2)(1+B^2)).
\end{aligned} \tag{79}$$

Moreover, applying Lemma 4.1 to (78) shows that

$$\mathbb{E}_{\text{init},\mu}\big[|\bar{\delta}(s,\boldsymbol{\theta}) - \widehat{\delta}(s,\boldsymbol{\theta})|\,|\widehat{\delta}(s,\boldsymbol{\theta})|\big] \leq \sqrt{3c_0(1+\bar{\rho}\gamma)(\bar{r}^2 + 2(1+\bar{\rho}^2\gamma^2)(1+B^2))B^3}\, m^{-1/4}.$$

Gathering terms yields

$$\begin{aligned}
&\mathbb{E}_{\text{init},\mu}[\bar{R}_\delta^{(1)}] \\
&\leq c_0(1+\bar{\rho}\gamma)B^3 m^{-1/2} + \sqrt{12c_0(1+\bar{\rho}\gamma)(\bar{r}^2 + 2(1+\bar{\rho}^2\gamma^2)(1+B^2))B^3}\, m^{-1/4} =: R_\delta^{(1)}.
\end{aligned}$$

We note that

$$R_f^{(1)} = \mathcal{O}(B^3 m^{-1/2} + B^{3/2} m^{-1/4}), \quad R_\delta^{(1)} = \mathcal{O}(B^3 m^{-1/2} + B^{3/2} m^{-1/4}).$$

To upper bound $\bar{R}_\delta^{(2)}$, we observe that

$$\begin{aligned}
\mathbb{E}_\mu\big[|\delta(s,\boldsymbol{\theta}) - \delta(s,\widehat{\boldsymbol{\theta}})|^2\big] &= \mathbb{E}_\mu\Big[\big|f(s,\boldsymbol{\theta}) - f(s,\widehat{\boldsymbol{\theta}}) - \gamma(f(s',\boldsymbol{\theta}) - f(s',\widehat{\boldsymbol{\theta}}))\big|^2\Big] \\
&\leq 3\,\mathbb{E}_\mu\Big[\big|\widehat{f}(s,\boldsymbol{\theta}) - \widehat{f}(s,\widehat{\boldsymbol{\theta}})\big|^2\Big] + 3\gamma^2\,\mathbb{E}_{\mu^\pi}\Big[\big|\widehat{f}(s',\boldsymbol{\theta}) - \widehat{f}(s',\widehat{\boldsymbol{\theta}})\big|^2\Big] + 12c_0 B^3 m^{-1/2} \\
&\leq 3(1+\bar{\rho}^2\gamma^2)d_{\widehat{f}}(\boldsymbol{z},\widehat{\boldsymbol{z}}) + 12c_0 B^3 m^{-1/2},
\end{aligned}$$

for $\boldsymbol{z} = (\boldsymbol{\theta}, \boldsymbol{w})$ with any $\boldsymbol{w} \in S_B$, where we have applied Lemma 4.1 in the second inequality. Moreover, using (79) and the definition of $\bar{R}_\delta^{(1)}$ we have

$$\mathbb{E}_{\text{init},\mu}\big[|\bar{\delta}(s,\widehat{\boldsymbol{\theta}})|^2\big] \leq R_\delta^{(1)} + 3(\bar{r}^2 + 2(1+\bar{\rho}^2\gamma^2)(1+B^2)).$$

The above shows that

$$\begin{aligned}
&\mathbb{E}_{\text{init},\mu}\Big[|\delta(s,\boldsymbol{\theta}) - \delta(s,\widehat{\boldsymbol{\theta}})|\,|\delta(s,\widehat{\boldsymbol{\theta}})|\Big] \leq \Big(\mathbb{E}_{\text{init},\mu}\big[|\delta(s,\widehat{\boldsymbol{\theta}})|^2\big]\Big)^{\frac{1}{2}} \Big(\mathbb{E}_{\text{init},\mu}\big[|\delta(s,\boldsymbol{\theta}) - \delta(s,\widehat{\boldsymbol{\theta}})|^2\big]\Big)^{\frac{1}{2}} \\
&\leq \sqrt{3(R_\delta^{(1)} + 3(\bar{r}^2 + 2(1+\bar{\rho}^2\gamma^2)(1+B^2)))}\sqrt{(1+\bar{\rho}^2\gamma^2)\mathbb{E}_{\text{init}}[d_{\widehat{f}}(\boldsymbol{z},\widehat{\boldsymbol{z}})] + 4c_0 B^3 m^{-1/2}}.
\end{aligned}$$

These yield

$$\begin{aligned}
&\mathbb{E}_{\text{init},\mu}[\bar{R}_\delta^{(2)}] \\
&\leq 3(1+\bar{\rho}^2\gamma^2)\mathbb{E}_{\text{init}}[d_{\widehat{f}}(\boldsymbol{z},\widehat{\boldsymbol{z}})] + 12c_0 B^3 m^{-1/2} \\
&+ \sqrt{12(R_\delta^{(1)} + 3(\bar{r}^2 + 2(1+\bar{\rho}^2\gamma^2)(1+B^2)))}\sqrt{(1+\bar{\rho}^2\gamma^2)\mathbb{E}_{\text{init}}[d_{\widehat{f}}(\boldsymbol{z},\widehat{\boldsymbol{z}})] + 4c_0 B^3 m^{-1/2}} =: R_\delta^{(2)}.
\end{aligned}$$

Finally, to upper bound $\bar{R}_f^{(2)}$, applying Lemma 4.1, we observe that

$$\mathbb{E}_\mu\big[|f(s,\boldsymbol{\theta}) - f(s,\widehat{\boldsymbol{\theta}})|^2\big] \leq 2\,\mathbb{E}_\mu\Big[\big|\widehat{f}(s,\boldsymbol{\theta}) - \widehat{f}(s,\widehat{\boldsymbol{\theta}})\big|^2\Big] + 4c_0 B^3 m^{-1/2} \leq 2d_{\widehat{f}}(\boldsymbol{z},\widehat{\boldsymbol{z}}) + 4c_0 B^3 m^{-1/2}.$$

Moreover,

$$\mathbb{E}_{\text{init},\mu}\big[|f(s,\widehat{\boldsymbol{\theta}})|^2\big] \leq R_f^{(1)} + 2(1+B^2).$$

This shows
$$\mathbb{E}_{\text{init},\mu}\big[|f(s,\widehat{\boldsymbol{\theta}})||f(s,\boldsymbol{\theta})-f(s,\widehat{\boldsymbol{\theta}})|\big] \leq \sqrt{2R_f^{(1)}+4(1+B^2)}\sqrt{\mathbb{E}_{\text{init}}[d_{\widehat{f}}(\boldsymbol{z},\widehat{\boldsymbol{z}})]+2c_0B^3m^{-1/2}}.$$

As such, we have

$$\mathbb{E}_{\text{init},\mu}[\bar{R}_f^{(2)}]$$

$$\leq 2\mathbb{E}_{\text{init}}[d_{\widehat{f}}(\boldsymbol{z},\widehat{\boldsymbol{z}})]+4c_0B^3m^{-1/2}+\sqrt{8R_f^{(1)}+16(1+B^2)}\sqrt{\mathbb{E}_{\text{init}}[d_{\widehat{f}}(\boldsymbol{z},\widehat{\boldsymbol{z}})]+2c_0B^3m^{-1/2}} =: R_f^{(2)}$$

We also note that

$$R_f^{(2)} = \mathcal{O}\left(\mathbb{E}_{\text{init}}[d_{\widehat{f}}(\boldsymbol{z},\widehat{\boldsymbol{z}})]+(B+B^{3/2}m^{-1/4})\sqrt{\mathbb{E}_{\text{init}}[d_{\widehat{f}}(\boldsymbol{z},\widehat{\boldsymbol{z}})]}+B^3m^{-1/2}+B^{5/2}m^{-1/4}\right),$$

$$R_\delta^{(2)} = \mathcal{O}\left(\mathbb{E}_{\text{init}}[d_{\widehat{f}}(\boldsymbol{z},\widehat{\boldsymbol{z}})]+(B+B^{3/2}m^{-1/4})\sqrt{\mathbb{E}_{\text{init}}[d_{\widehat{f}}(\boldsymbol{z},\widehat{\boldsymbol{z}})]}+B^3m^{-1/2}+B^{5/2}m^{-1/4}\right).$$

Note that $R_f^{(0)}, R_f^{(1)}, R_f^{(2)}, R_\delta^{(0)}, R_\delta^{(1)}, R_\delta^{(2)}$ are deterministic quantities.

**Bounding $J_\upsilon(\boldsymbol{\theta})$**   In the following derivations, the inequalities hold in the expectation taken over the NN initializations. We shall skip $\mathbb{E}_{\text{init}}[\cdot]$ to simplify notations. First observe the following bound using the above definitions

$$J_\upsilon(\boldsymbol{\theta}) \leq J_\upsilon(\widehat{\boldsymbol{\theta}})+\frac{1}{2}\left(R_\delta^{(2)}+\upsilon R_f^{(2)}\right) \tag{80}$$

With a slight abuse in notations, define the following function from the Fenchel's conjugate (8):

$$\overline{J}_\upsilon(\boldsymbol{\theta};y(\cdot)) := \mathbb{E}_\mu\left[y(s)\bar{\delta}(s,\boldsymbol{\theta})-\frac{1}{2}y(s)^2+\frac{\upsilon}{2}f(s,\boldsymbol{\theta})^2\right] \tag{81}$$

From H4, there exists $\boldsymbol{w}(\widehat{\boldsymbol{\theta}}) \in S_B$ such that $\mathbb{E}_\mu[|\bar{\delta}(s,\widehat{\boldsymbol{\theta}})-f(s,\boldsymbol{w}(\widehat{\boldsymbol{\theta}}))|^2] \leq c_{\text{nn}}$. As such, we observe that

$$J_\upsilon(\widehat{\boldsymbol{\theta}}) = \overline{J}_\upsilon(\widehat{\boldsymbol{\theta}};\bar{\delta}(\cdot,\widehat{\boldsymbol{\theta}}))$$

$$= J_\upsilon(\widehat{\boldsymbol{\theta}},\boldsymbol{w}(\widehat{\boldsymbol{\theta}}))+\mathbb{E}_\mu\left[\bar{\delta}(s,\widehat{\boldsymbol{\theta}})(\bar{\delta}(s,\widehat{\boldsymbol{\theta}})-f(s,\boldsymbol{w}(\widehat{\boldsymbol{\theta}})))-\frac{1}{2}\{\bar{\delta}(s,\widehat{\boldsymbol{\theta}})^2-f(s,\boldsymbol{w}(\widehat{\boldsymbol{\theta}}))^2\}\right] \tag{82}$$

$$= J_\upsilon(\widehat{\boldsymbol{\theta}},\boldsymbol{w}(\widehat{\boldsymbol{\theta}}))+\frac{1}{2}\mathbb{E}_\mu\left[|f(s,\boldsymbol{w}(\widehat{\boldsymbol{\theta}}))-\bar{\delta}(s,\widehat{\boldsymbol{\theta}})|^2\right] \leq J_\upsilon(\widehat{\boldsymbol{\theta}},\boldsymbol{w}(\widehat{\boldsymbol{\theta}}))+\frac{c_{\text{nn}}}{2}.$$

Using the bounds developed earlier in this subsection, we have

$$J_\upsilon(\widehat{\boldsymbol{\theta}},\boldsymbol{w}(\widehat{\boldsymbol{\theta}})) \leq \widehat{J}_\upsilon(\widehat{\boldsymbol{\theta}},\boldsymbol{w}(\widehat{\boldsymbol{\theta}}))+\frac{1}{2}(1+\upsilon)R_f^{(1)}$$

$$+\frac{1}{2}\mathbb{E}_\mu\left[|\bar{\delta}(s,\widehat{\boldsymbol{\theta}})||f(s,\boldsymbol{w}(\widehat{\boldsymbol{\theta}}))-\widehat{f}(s,\boldsymbol{w}(\widehat{\boldsymbol{\theta}}))|+|\widehat{f}(s,\boldsymbol{w}(\widehat{\boldsymbol{\theta}}))||\bar{\delta}(s,\widehat{\boldsymbol{\theta}})-\widehat{\delta}(s,\widehat{\boldsymbol{\theta}})|\right].$$

Since $(\widehat{\boldsymbol{\theta}},\widehat{\boldsymbol{w}})$ is a saddle point to (15), we have

$$\widehat{J}_\upsilon(\widehat{\boldsymbol{\theta}},\boldsymbol{w}(\widehat{\boldsymbol{\theta}})) \leq \widehat{J}_\upsilon(\widehat{\boldsymbol{\theta}},\widehat{\boldsymbol{w}}) \leq \widehat{J}_\upsilon(\boldsymbol{\theta}^\star,\widehat{\boldsymbol{w}}) \tag{83}$$

Notice that

$$\widehat{J}_\upsilon(\boldsymbol{\theta}^\star,\widehat{\boldsymbol{w}}) \leq J_\upsilon(\boldsymbol{\theta}^\star,\widehat{\boldsymbol{w}})+\frac{1}{2}(1+\upsilon)R_f^{(1)}$$

$$+\frac{1}{2}\mathbb{E}_\mu\left[|\widehat{\delta}(s,\boldsymbol{\theta}^\star)||f(s,\widehat{\boldsymbol{w}})-\widehat{f}(s,\widehat{\boldsymbol{w}})|+|f(s,\widehat{\boldsymbol{w}})||\bar{\delta}(s,\boldsymbol{\theta}^\star)-\widehat{\delta}(s,\boldsymbol{\theta}^\star)|\right].$$

Finally, we observe,

$$J_\upsilon(\boldsymbol{\theta}^\star,\widehat{\boldsymbol{w}}) = \overline{J}_\upsilon(\boldsymbol{\theta}^\star;f(\cdot,\widehat{\boldsymbol{w}})) \leq \overline{J}_\upsilon(\boldsymbol{\theta}^\star;\bar{\delta}(\cdot,\boldsymbol{\theta}^\star)) = J_\upsilon(\boldsymbol{\theta}^\star),$$

where the inequality is due to the fact that $\bar{\bar{\delta}}(\cdot, \boldsymbol{\theta}^\star)$ maximizes $\overline{J}_v(\boldsymbol{\theta}^\star; y(\cdot))$. Collecting terms, we observe that

$$
\begin{aligned}
&J_v(\boldsymbol{\theta}) - J_v(\boldsymbol{\theta}^\star) - \frac{1}{2}\big(\mathsf{c}_{\mathsf{nn}} + R_\delta^{(2)} + v R_f^{(2)}\big) \\
&\leq (1+v)R_f^{(1)} \\
&+ \frac{1}{2}\mathbb{E}_\mu\left[|\bar{\delta}(s,\widehat{\boldsymbol{\theta}})||f(s,\boldsymbol{w}(\widehat{\boldsymbol{\theta}})) - \widehat{f}(s,\boldsymbol{w}(\widehat{\boldsymbol{\theta}}))| + |\widehat{f}(s,\boldsymbol{w}(\widehat{\boldsymbol{\theta}}))||\bar{\delta}(s,\widehat{\boldsymbol{\theta}}) - \widehat{\delta}(s,\widehat{\boldsymbol{\theta}})|\right] \\
&+ \frac{1}{2}\mathbb{E}_\mu\left[|\widehat{\delta}(s,\boldsymbol{\theta}^\star)||f(s,\widehat{\boldsymbol{w}}) - \widehat{f}(s,\widehat{\boldsymbol{w}})| + |f(s,\widehat{\boldsymbol{w}})||\bar{\delta}(s,\boldsymbol{\theta}^\star) - \widehat{\delta}(s,\boldsymbol{\theta}^\star)|\right] \\
&\leq \frac{1}{2}\left(\sqrt{R_\delta^{(1)} + 3(\bar{r}^2 + 2(1+\bar{\rho}^2\gamma^2)(1+B^2))}\sqrt{R_f^{(0)}} + \sqrt{2(1+B^2)}\sqrt{R_\delta^{(0)}}\right) \\
&+ \frac{1}{2}\left(\sqrt{3(\bar{r}^2 + 2(1+\bar{\rho}^2\gamma^2)(1+B^2))}\sqrt{R_f^{(0)}} + \sqrt{R_f^{(1)} + 2(1+B^2)}\sqrt{R_\delta^{(0)}}\right)
\end{aligned} \tag{84}
$$

Finally, we obtain the following bound

$$
\begin{aligned}
&J_v(\boldsymbol{\theta}) - J_v(\boldsymbol{\theta}^\star) \\
&\leq \frac{1}{2}\big(\mathsf{c}_{\mathsf{nn}} + R_\delta^{(2)} + v R_f^{(2)}\big) + (1+v)R_f^{(1)} \\
&+ \frac{1}{2}\left(\sqrt{R_\delta^{(1)} + 3(\bar{r}^2 + 2(1+\bar{\rho}^2\gamma^2)(1+B^2))}\sqrt{R_f^{(0)}} + \sqrt{2(1+B^2)}\sqrt{R_\delta^{(0)}}\right) \\
&+ \frac{1}{2}\left(\sqrt{3(\bar{r}^2 + 2(1+\bar{\rho}^2\gamma^2)(1+B^2))}\sqrt{R_f^{(0)}} + \sqrt{R_f^{(1)} + 2(1+B^2)}\sqrt{R_\delta^{(0)}}\right)
\end{aligned} \tag{85}
$$

The proof is completed by extracting the right orders in terms of $m, B, \mathbb{E}_{\mathsf{init}}[d_{\widehat{f}}(\boldsymbol{z}, \widehat{\boldsymbol{z}})]$ from the previously derived bounds of $R_f^{(0)}, R_f^{(1)}, R_f^{(2)}, R_\delta^{(0)}, R_\delta^{(1)}, R_\delta^{(2)}$.

## D   Convergence of Neural GTD Algorithm with Markov Samples

We consider a more sophisticated setting when samples used for the neural GTD algorithm are collected along a single sample path of the Markov chain $(s_0, s_1, ...)$. At iteration $k$ of the neural GTD algorithm, the state pair is taken as $\widetilde{s}_k := (s_k, s_{k+1})$, see Algorithm 1. In this case, the neural GTD update can be written into the same form as (21), (22). However, for finite $k$, as $(\widetilde{s}_k)_{k \geq 0}$ is not drawn from the stationary distribution, denoted as $\widetilde{\mu}^\Pi$, H3 does not hold and the analysis leading to Theorem 3.2 cannot be applied. Instead, $(\widetilde{s}_k)_{k \geq 0}$ forms a uniformly geometric ergodic Markov chain whose kernel is denoted by $\widetilde{\mathrm{P}}^\Pi$. Let us define $\rho \in [0, 1)$ be its mixing constant. Since the Markov chain is uniformly geometric ergodic, there exists a constant $\mathrm{K}_P$ such that

$$
\sup_{\widetilde{s} \in \mathsf{S} \times \mathsf{S}} \|(\widetilde{\mathrm{P}}^\Pi)^n(\widetilde{s}, \cdot) - \widetilde{\mu}^\Pi(\cdot)\| \leq \rho^n \mathrm{K}_P, \tag{86}
$$

where $\| \cdot \|$ denotes the total variation norm.

We use the standard Bachmann-Landau notations for asymptotic quantities in the following. In particular, consider two non-negative functions $h(x), g(x)$. We say that $h(x) = \mathcal{O}(g(x))$ if there exists $c_0 > 0, x_0 \geq 0$ such that $h(x) \leq c_0 g(x)$ for all $x \geq x_0$; likewise, $h(x) = \Omega(g(x))$ if there exists $c_1 > 0, x_1 \geq 0$ such that $h(x) \geq c_1 g(x)$ for all $x \geq x_0$.

As explained in the main text of the paper, in the Markov sample settings, we prove the convergence of neural GTD *with high probability* with respect to the random initialization specified in H1. To this end, we need to derive the high probability bounds for certain quantities. Observe the lemmas:

**Lemma D.1.** *Assume that H1 holds, $m = \Omega(d^{3/2})$, and $B = \mathcal{O}(m^{1/2}(\log m)^{-3})$. For any $(\boldsymbol{\theta}, \boldsymbol{w}) \in S_B \times S_B$, and $\widetilde{s} \in \mathsf{S} \times \mathsf{S}$, with probability at least $1 - e^{\Omega(\log^2 m)}$ with respect to the random initialization, it holds*

$$
\|\nabla_{\boldsymbol{\theta}} J_v(\boldsymbol{\theta}, \boldsymbol{w}; \widetilde{s})\|_2 \vee \|\nabla_{\boldsymbol{w}} J_v(\boldsymbol{\theta}, \boldsymbol{w}; \widetilde{s})\|_2 \leq \mathcal{O}(B \vee \log m) =: \varsigma. \tag{87}
$$

**Lemma D.2.** *Assume that H1 holds, $m = \Omega(d^{3/2})$, and $B = \mathcal{O}(m^{1/2}(\log m)^{-3})$. For any $\boldsymbol{z}, \boldsymbol{z}' \in S_B \times S_B$ and $\widetilde{s} \in \mathsf{S} \times \mathsf{S}$, there exist constants $\mathrm{L}_J, \mathrm{C}_J$ such that with probability at least $1 - e^{\log^2(m)}$ with respect to the random initialization,*

$$
\|\nabla J_v(\boldsymbol{z}; \widetilde{s}) - \nabla J_v(\boldsymbol{z}'; \widetilde{s})\|_2 \leq \mathrm{L}_J \|\boldsymbol{z} - \boldsymbol{z}'\|_2 + \mathrm{C}_J B^{4/3}(\log m)^{3/2} m^{-1/6}. \tag{88}
$$

**Lemma D.3.** *Assume that H1 holds, $m = \Omega(d^{3/2})$, and $B = \mathcal{O}(m^{1/2}(\log m)^{-3})$. For any $\boldsymbol{\theta} \in S_B$ and $s \in \mathsf{S}$, with probability at least $1 - e^{\Omega(B^{2/3}m^{2/3})}$ with respect to the random initialization, it holds*

$$|f(s, \boldsymbol{\theta}) - \widehat{f}(s, \boldsymbol{\theta})| = \mathcal{O}(B^{4/3}m^{-1/6}\sqrt{\log m}). \tag{89}$$

We present the main conclusion on the convergence of neural GTD in detail [cf. Theorem 3.3] below:

**Theorem D.1.** *Assume that H1 hold, $m = \Omega(d^{3/2})$, and $B = \mathcal{O}(m^{1/2}(\log m)^{-3})$. Furthermore, for any $k \geq 0$, there exists a constant $\xi$ such that*

$$|\beta_k - \beta_{k+1}| \leq \xi\beta_k^2. \tag{90}$$

*For the iterates generated by Algorithm 1 with the Markov chain satisfying (86), and any $n \geq 1$, it holds with probability at least $1 - e^{\Omega(\log^2 m)}$ over the random initialization that:*

$$\mathbb{E}_{I_n}\left[d_{\widehat{f}}(\boldsymbol{z}_{I_n}, \widehat{\boldsymbol{z}})\right] = \mathcal{O}\left(\frac{B^{\frac{8}{3}} + B^{\frac{7}{3}}(\log m)^{\frac{1}{2}}/(1-\rho)}{(1 \wedge \upsilon)m^{\frac{1}{6}}/\log m} + \frac{\|\boldsymbol{z}_0 - \widehat{\boldsymbol{z}}\|_2^2 + \beta_0\frac{B^2}{1-\rho} + (B^3 + \frac{B^2}{1-\rho})\sum_{k=0}^n \beta_k^2}{(1 \wedge \upsilon)\sum_{k=0}^n \beta_k}\right), \tag{91}$$

*where the function $d_{\widehat{f}}(\boldsymbol{z}_{I_n}, \widehat{\boldsymbol{z}})$ was defined in (18). Moreover, the expectation above is taken over the independent r.v. $I_n$, and the i.i.d. samples of states drawn during the algorithm.*

Lastly, we remark that the conditions $m = \Omega(d^{3/2})$, and $B = \mathcal{O}(m^{1/2}(\log m)^{-3})$ can be satisfied when $m$ is sufficiently large and we choose $B = \mathcal{O}(1)$.

*Proof.* Now, let us recall that the neural GTD update can be written as:

$$\begin{aligned}
\boldsymbol{\theta}_{k+1} &= \mathcal{P}_{S_B}\left\{\boldsymbol{\theta}_k - \beta_k\left(\nabla_{\boldsymbol{\theta}}\widehat{J}_{\upsilon}(\boldsymbol{z}_k) + \widehat{\boldsymbol{e}}_k^{(1)} + \boldsymbol{e}_k^{(1)}\right)\right\}, \\
\boldsymbol{w}_{k+1} &= \mathcal{P}_{S_B}\left\{\boldsymbol{w}_k + \beta_k\left(\nabla_{\boldsymbol{w}}\widehat{J}_{\upsilon}(\boldsymbol{z}_k) + \widehat{\boldsymbol{e}}_k^{(2)} + \boldsymbol{e}_k^{(2)}\right)\right\},
\end{aligned} \tag{92}$$

and that the errors are expressed as:

$$\begin{aligned}
\widehat{\boldsymbol{e}}_k^{(1)} &= \nabla_{\boldsymbol{\theta}}J_{\upsilon}(\boldsymbol{z}_k) - \nabla_{\boldsymbol{\theta}}\widehat{J}_{\upsilon}(\boldsymbol{z}_k), \quad \boldsymbol{e}_k^{(1)} = \nabla_{\boldsymbol{\theta}}J_{\upsilon}(\boldsymbol{z}_k; \widetilde{s}_k) - \mathbb{E}_{\widetilde{\mu}}[\nabla_{\boldsymbol{\theta}}J_{\upsilon}(\boldsymbol{z}_k; \widetilde{s})]. \\
\widehat{\boldsymbol{e}}_k^{(2)} &= \nabla_{\boldsymbol{w}}J_{\upsilon}(\boldsymbol{z}_k) - \nabla_{\boldsymbol{w}}\widehat{J}_{\upsilon}(\boldsymbol{z}_k), \quad \boldsymbol{e}_k^{(2)} = \nabla_{\boldsymbol{w}}J_{\upsilon}(\boldsymbol{z}_k; \widetilde{s}_k) - \mathbb{E}_{\widetilde{\mu}}[\nabla_{\boldsymbol{w}}J_{\upsilon}(\boldsymbol{z}_k; \widetilde{s})],
\end{aligned} \tag{93}$$

where the expectation $\mathbb{E}_{\widetilde{\mu}}[\cdot]$ is evaluated with respect to the stationary distribution $\widetilde{\mu}^{\Pi}$ of the Markov chain $(\widetilde{s}_k)_{k \geq 0}$. We observe the following lemma which is analogous to Lemma A.1:

**Lemma D.4.** *Assume that H1, H2 hold. For any $\boldsymbol{z} \in S_B \times S_B$, with probability at least $1 - e^{\Omega(\log^2 m)}$ with respect to the random initialization, it holds*

$$\|\boldsymbol{e}_k^{(1)}\|_2^2 \vee \|\boldsymbol{e}_k^{(2)}\|_2^2 \leq \mathrm{C}^e B^{8/3}m^{-1/3}(\log m)^2. \tag{94}$$

Using [Douc et al., 2018, Proposition 21.2.3], together with Lemma D.1 and some regulatory conditions, there exists measurable functions $\widehat{\mathrm{dJ}}_{\upsilon}^{(i)} : S_B \times \mathsf{S} \to \mathbb{R}^{md}$, $i = 1, 2$, satisfying

$$\boldsymbol{e}_k^{(i)} = \widehat{\mathrm{dJ}}_{\upsilon}^{(i)}(\boldsymbol{z}_k; \widetilde{s}_k) - \widetilde{\mathrm{P}}^{\Pi}\widehat{\mathrm{dJ}}_{\upsilon}^{(i)}(\boldsymbol{z}_k; \widetilde{s}_k), \ i = 1, 2. \tag{95}$$

This is also known as the Poisson equation. Again with Lemma D.1, it can be shown that (e.g., using [Fort et al., 2011])

$$\sup_{\boldsymbol{z} \in S_B \times S_B, \widetilde{s} \in \mathsf{S} \times \mathsf{S}} \|\widehat{\mathrm{dJ}}_{\upsilon}^{(i)}(\boldsymbol{z}; \widetilde{s})\|_2 \leq \varsigma\, \mathrm{K}_P/(1-\rho) =: \widehat{\varsigma},$$

Furthermore, for any $\widetilde{s} \in \mathsf{S} \times \mathsf{S}$ and $\boldsymbol{z}, \boldsymbol{z}' \in S_B \times S_B$, using Lemma D.2,

$$\|\widehat{\mathrm{dJ}}_{\upsilon}^{(i)}(\boldsymbol{z}; \widetilde{s}) - \widehat{\mathrm{dJ}}_{\upsilon}^{(i)}(\boldsymbol{z}'; \widetilde{s})\|_2 \leq \mathrm{L}_J \mathrm{K}_P/(1-\rho)\|\boldsymbol{z} - \boldsymbol{z}'\|_2 + (\mathrm{C}_J \mathrm{K}_P/(1-\rho))B^{4/3}(\log m)^{3/2}m^{-1/6}, \tag{96}$$

for brevity, we denote $\widehat{\mathrm{L}}_J := \mathrm{L}_J \mathrm{K}_P/(1-\rho)$, $\widehat{\mathrm{C}}_J := \mathrm{C}_J \mathrm{K}_P/(1-\rho)$.

We proceed to proving the convergence of neural GTD by following the analysis done in previous sections. Since the neural GTD update follows the same form as in (21), (22), we observe that every steps in the proof of Theorem 3.2 follows. Also, the second inequality in H3 holds with $\sigma^2$ replaced by $4\varsigma^2$. In particular, applying Lemma A.2 and we observe that the following (which is analogous to (71)) holds for any $k \geq 0$, $c_1 > 0$

$$\mathbb{E}[\|\boldsymbol{z}_{k+1} - \widehat{\boldsymbol{z}}\|_2^2 | \mathcal{F}_{k-1}] \leq \|\boldsymbol{z}_k - \widehat{\boldsymbol{z}}\|_2^2 - 2\beta_k \mu d_{\widehat{f}}(\boldsymbol{z}_k, \widehat{\boldsymbol{z}}) + 3\beta_k^2 L_\Phi d_{\widehat{f}}(\boldsymbol{z}_k, \widehat{\boldsymbol{z}}) + \frac{\beta_k}{c_1} 4B^2$$

$$+ \left(3\beta_k^2 + c_1\beta_k\right)\left\{\|\widehat{\boldsymbol{e}}_k^{(1)}\|_2^2 + \|\widehat{\boldsymbol{e}}_k^{(2)}\|_2^2\right\} + 24\beta_k^2\varsigma^2$$

$$- 2\beta_k \mathbb{E}\left[\langle \boldsymbol{\theta}_k - \widehat{\boldsymbol{\theta}}, \boldsymbol{e}_k^{(1)}\rangle - \langle \boldsymbol{w}_k - \widehat{\boldsymbol{w}}, \boldsymbol{e}_k^{(2)}\rangle | \mathcal{F}_{k-1}\right],$$

where the last term is new. Using the step size condition, Lemma D.4 and setting $c_1 = m^{1/6}/\log m$, we can derive the following inequality:

$$\mathbb{E}\left[d_{\widehat{f}}(\boldsymbol{z}_{I_n}, \widehat{\boldsymbol{z}})\right] \leq \frac{2C^e B^{\frac{8}{3}} + 4B^2}{\mu m^{\frac{1}{6}}/\log m} + \frac{\mathbb{E}[\|\boldsymbol{z}_0 - \widehat{\boldsymbol{z}}\|_2^2] + 2\left\{3C^e B^{\frac{8}{3}} m^{-1/3}(\log m)^2 + 12\varsigma^2\right\} \sum_{k=0}^n \beta_k^2}{\mu \sum_{k=0}^n \beta_k}$$

$$+ \frac{2\left|\sum_{k=0}^n \beta_k \mathbb{E}\left[\langle \boldsymbol{\theta}_k - \widehat{\boldsymbol{\theta}}, \boldsymbol{e}_k^{(1)}\rangle - \langle \boldsymbol{w}_k - \widehat{\boldsymbol{w}}, \boldsymbol{e}_k^{(2)}\rangle\right]\right|}{\mu \sum_{k=0}^n \beta_k}. \tag{97}$$

Our remaining task is to bound the weighted sum of inner products in the last term. Observe

$$\beta_k \left\langle \boldsymbol{\theta}_k - \widehat{\boldsymbol{\theta}}, \boldsymbol{e}_k^{(1)} \right\rangle = \beta_k \left\langle \boldsymbol{\theta}_k - \widehat{\boldsymbol{\theta}}, \widehat{\mathrm{dJ}}_v^{(1)}(\boldsymbol{z}_k; \widetilde{s}_k) - \widehat{\mathrm{dJ}}_v^{(1)}(\boldsymbol{z}_k; \widetilde{s}_{k+1}) + \widehat{\mathrm{dJ}}_v^{(1)}(\boldsymbol{z}_k; \widetilde{s}_{k+1}) - \widetilde{\mathrm{P}}^\Pi \widehat{\mathrm{dJ}}_v^{(1)}(\boldsymbol{z}_k; \widetilde{s}_k) \right\rangle.$$

By (95), we observe that $\widehat{\mathrm{dJ}}_v^{(1)}(\boldsymbol{z}_k; \widetilde{s}_{k+1}) - \widetilde{\mathrm{P}}^\Pi \widehat{\mathrm{dJ}}_v^{(1)}(\boldsymbol{z}_k; \widetilde{s}_k)$ is a martingale difference such that $\mathbb{E}[\langle \boldsymbol{\theta}_k - \widehat{\boldsymbol{\theta}}, \widehat{\mathrm{dJ}}_v^{(1)}(\boldsymbol{z}_k; \widetilde{s}_{k+1}) - \widetilde{\mathrm{P}}^\Pi \widehat{\mathrm{dJ}}_v^{(1)}(\boldsymbol{z}_k; \widetilde{s}_k)\rangle | \mathcal{F}_{k-1}] = 0$, where we recall $\mathcal{F}_k = \sigma\{\boldsymbol{\theta}_0, \widetilde{s}_0, \widetilde{s}_1, ..., \widetilde{s}_k\}$. Furthermore, the remaining inner product can be decomposed into four terms:

$$\beta_k \left\langle \boldsymbol{\theta}_k - \widehat{\boldsymbol{\theta}}, \widehat{\mathrm{dJ}}_v^{(1)}(\boldsymbol{z}_k; \widetilde{s}_k) - \widehat{\mathrm{dJ}}_v^{(1)}(\boldsymbol{z}_k; \widetilde{s}_{k+1}) \right\rangle$$

$$= \underbrace{\beta_k \left\langle \boldsymbol{\theta}_k - \widehat{\boldsymbol{\theta}}, \widehat{\mathrm{dJ}}_v^{(1)}(\boldsymbol{z}_k; \widetilde{s}_k) \right\rangle - \beta_{k+1} \left\langle \boldsymbol{\theta}_{k+1} - \widehat{\boldsymbol{\theta}}, \widehat{\mathrm{dJ}}_v^{(1)}(\boldsymbol{z}_{k+1}; \widetilde{s}_{k+1}) \right\rangle}_{=A_k^1}$$

$$+ \underbrace{(\beta_{k+1} - \beta_k) \left\langle \boldsymbol{\theta}_{k+1} - \widehat{\boldsymbol{\theta}}, \widehat{\mathrm{dJ}}_v^{(1)}(\boldsymbol{z}_{k+1}; \widetilde{s}_{k+1}) \right\rangle}_{=A_k^2} + \underbrace{\beta_k \left\langle \boldsymbol{\theta}_{k+1} - \boldsymbol{\theta}_k, \widehat{\mathrm{dJ}}_v^{(1)}(\boldsymbol{z}_{k+1}; \widetilde{s}_{k+1}) \right\rangle}_{=A_k^3}$$

$$+ \underbrace{\beta_k \left\langle \boldsymbol{\theta}_k - \widehat{\boldsymbol{\theta}}, \widehat{\mathrm{dJ}}_v^{(1)}(\boldsymbol{z}_{k+1}; \widetilde{s}_{k+1}) - \widehat{\mathrm{dJ}}_v^{(1)}(\boldsymbol{z}_k; \widetilde{s}_{k+1}) \right\rangle}_{=A_k^4}.$$

In particular, we have

$$\left|\sum_{k=0}^n A_k^1\right| = \beta_0 \left\langle \boldsymbol{\theta}_0 - \widehat{\boldsymbol{\theta}}, \widehat{\mathrm{dJ}}_v^{(1)}(\boldsymbol{z}_0; \widetilde{s}_0) \right\rangle - \beta_{n+1} \left\langle \boldsymbol{\theta}_{n+1} - \widehat{\boldsymbol{\theta}}, \widehat{\mathrm{dJ}}_v^{(1)}(\boldsymbol{z}_{n+1}; \widetilde{s}_{n+1}) \right\rangle \leq (\beta_0 + \beta_{n+1}) B\widehat{\varsigma},$$

$$\left|\sum_{k=0}^n A_k^2\right| \overset{(a)}{\leq} \xi B\widehat{\varsigma} \sum_{k=0}^n \beta_k^2, \qquad \left|\sum_{k=0}^n A_k^3\right| \overset{(b)}{\leq} \widehat{\varsigma}\varsigma \sum_{k=0}^n \beta_k^2,$$

$$\left|\sum_{k=0}^n A_k^4\right| \overset{(c)}{\leq} 2\widehat{\mathrm{L}}_J B\varsigma \sum_{k=0}^n \beta_k^2 + \widehat{\mathrm{C}}_J \frac{B^{7/3}(\log m)^{3/2}}{m^{1/6}} \sum_{k=0}^n \beta_k,$$

where (a) is due to the additional condition (90) on the step size $\beta_k$; (b) is due to the uniform boundedness on sampled gradient, i.e., Lemma D.1, as well as the non-expansive property of

projection; (c) is due to the Lipschitz gradient condition in Lemma D.2 and (96). Notice that the same analysis is applicable to the sum of inner product $\beta_k \langle \boldsymbol{w}_k - \widehat{\boldsymbol{w}}, \boldsymbol{e}_k^{(1)} \rangle$. The above analysis shows that

$$
\begin{aligned}
&2 \left| \sum_{k=0}^{n} \beta_k \mathbb{E}\big[ \langle \boldsymbol{\theta}_k - \widehat{\boldsymbol{\theta}}, \boldsymbol{e}_k^{(1)} \rangle - \langle \boldsymbol{w}_k - \widehat{\boldsymbol{w}}, \boldsymbol{e}_k^{(2)} \rangle \big] \right| \\
&\leq 4 \Big\{ (\beta_0 + \beta_{n+1}) B\widehat{\varsigma} + (\xi B\widehat{\varsigma} + \widehat{\varsigma}\varsigma + 2\widehat{\mathrm{L}}_J B\varsigma) \sum_{k=0}^{n} \beta_k^2 + \widehat{\mathrm{C}}_J \frac{B^{7/3}(\log m)^{3/2}}{m^{1/6}} \sum_{k=0}^{n} \beta_k \Big\}.
\end{aligned}
\tag{98}
$$

Plugging the above into (97) gives the convergence rate as

$$
\begin{aligned}
\mathbb{E}\big[ d_f(\boldsymbol{z}_{I_n}, \boldsymbol{z}^\star) \big] &\leq \frac{2\mathrm{C}^e B^{\frac{8}{3}} + 4B^2}{\mu m^{\frac{1}{6}}/\log m} + \frac{\mathbb{E}[\|\boldsymbol{z}_0 - \widehat{\boldsymbol{z}}\|_2^2] + 2\big\{ 3\mathrm{C}^e B^{\frac{8}{3}} m^{-1/3}(\log m)^2 + 12\varsigma^2 \big\} \sum_{k=0}^{n} \beta_k^2}{\mu \sum_{k=0}^{n} \beta_k} \\
&\quad + \frac{2(\beta_0 + \beta_{n+1}) B\widehat{\varsigma} + 2(\xi B\widehat{\varsigma} + \widehat{\varsigma}\varsigma + 2\widehat{\mathrm{L}}_J B\varsigma) \sum_{k=0}^{n} \beta_k^2}{\mu \sum_{k=0}^{n} \beta_k} + \frac{4\widehat{\mathrm{C}}_J B^{7/3}(\log m)^{3/2}}{m^{1/6}} \\
&= \mathcal{O}\left( \frac{B^{\frac{8}{3}} + B^{\frac{7}{3}}(\log m)^{\frac{1}{2}}/(1-\rho)}{(1 \wedge \upsilon) m^{\frac{1}{6}}/\log m} + \frac{\|\boldsymbol{z}_0 - \widehat{\boldsymbol{z}}\|_2^2 + \beta_0 \frac{B^2}{1-\rho} + \frac{B^2}{1-\rho} \sum_{k=0}^{n} \beta_k^2}{(1 \wedge \upsilon) \sum_{k=0}^{n} \beta_k} \right),
\end{aligned}
\tag{99}
$$

where we have recalled that $\mu = 1 \wedge \upsilon$ and assumed $B = \Omega(\log m)$ in estimating the upper bound for $\varsigma, \widehat{\varsigma}$.

We thus conclude that the neural GTD algorithm converges when using Markov samples, yet the rate holds with high probability with respect to the random initialization. Moreover, the asymptotic bias of $\mathcal{O}(B^{\frac{8}{3}}(\log m)^{\frac{3}{2}}/((1-\rho)(1 \wedge \upsilon)m^{\frac{1}{6}}))$ is higher than that of the i.i.d. case. $\qquad \square$

Lastly, we present the following result that is akin to Theorem 3.4, but holds with high probability with respect to the random initialization of NN.

**Corollary D.1.** *Assume H1, H4. For any $\boldsymbol{\theta} \in S_B$, with probability at least $1 - e^{\Omega(\log^2 m)}$, it holds*

$$
\begin{aligned}
&J_\upsilon(\boldsymbol{\theta}) - J_\upsilon(\boldsymbol{\theta}^\star) \\
&\leq \mathcal{O}\left( \mathrm{c}_{\mathsf{nn}} + d_{\widehat{f}}(\boldsymbol{z}, \widehat{\boldsymbol{z}}) + B\sqrt{d_{\widehat{f}}(\boldsymbol{z}, \widehat{\boldsymbol{z}})} + B^{8/3} m^{-1/3} \log m + B^{7/3} m^{-1/6} \sqrt{\log m} \right),
\end{aligned}
$$

*where $\boldsymbol{z}$ is defined as the vector $\boldsymbol{z} = (\boldsymbol{\theta}, \boldsymbol{w})$ for any $\boldsymbol{w} \in S_B$, and $d_{\widehat{f}}(\boldsymbol{z}, \widehat{\boldsymbol{z}}(\Xi_0))$ was defined in (18).*

*Proof.* The proof follows from repeating the steps from the previous section using the high probability bounds in this appendix. Following the previous notations, it can be shown that it holds with probability at least $1 - e^{\Omega(\log^2 m)}$ for

$$
\text{r.h.s. of (84)} = \mathcal{O}\left( B^{8/3} m^{-1/3} \log m + B^{7/3} m^{-1/6} \sqrt{\log m} \right),
$$

where we have repeatedly applied Lemma D.3 and [Allen-Zhu et al., 2019b, Theorem 1], see (101).

Moreover, for $R_\delta^{(2)}, R_f^{(2)}$, it can be shown that

$$
R_\delta^{(2)} \vee R_f^{(2)} = \mathcal{O}\left( d_{\widehat{f}}(\boldsymbol{z}, \widehat{\boldsymbol{z}}) + B\sqrt{d_{\widehat{f}}(\boldsymbol{z}, \widehat{\boldsymbol{z}})} \right).
$$

Combining the terms yields the desired corollary. $\qquad \square$

## D.1 Proof of Lemma D.1

First we observe that the gradient is given by:

$$
\nabla_{\boldsymbol{\theta}} J_\upsilon(\boldsymbol{\theta}, \boldsymbol{w}; \widetilde{s}) = f(s, \boldsymbol{w}) \nabla \delta(s, a, s'; \boldsymbol{\theta}) + \upsilon f(s, \boldsymbol{\theta}) \nabla f(s, \boldsymbol{\theta}),
\tag{100}
$$

where we recall the definition of $\delta(\cdot)$ from (5). Using [Gao et al., 2019, Lemma A.5], it can be shown that with probability at least $1 - e^{\Omega(B^{2/3} m^{2/3})}$, we have for any $s, a, s' \in \mathsf{S} \times \mathsf{A} \times \mathsf{S}$,

$$
\|\nabla f(s, \boldsymbol{\theta})\|_2 = \mathcal{O}(1), \quad \|\nabla \delta(s, a, s'; \boldsymbol{\theta})\|_2 = \mathcal{O}(1),
$$

where we have also used $\sup_{a,s} \rho(a|s) = \mathcal{O}(1)$. Furthermore, from [Allen-Zhu et al., 2019b, Theorem 1], with probability at least $1 - e^{\Omega(\log^2 m)}$, we have

$$|f(s, \boldsymbol{\theta})| \leq |f(s, \boldsymbol{\theta}_0)| + |f(s, \boldsymbol{\theta}) - f(s, \boldsymbol{\theta}_0)| = \mathcal{O}(B \vee \log m), \tag{101}$$

where we have used the Lipschitz property of the NN function where $|f(s, \boldsymbol{\theta}) - f(s, \boldsymbol{\theta}_0)| = \mathcal{O}(B)$. Combining the above observations gives $\|\nabla_{\boldsymbol{\theta}} J_\upsilon(\boldsymbol{\theta}, \boldsymbol{w}; \widetilde{s})\|_2 = \mathcal{O}(B \vee \log m)$ with probability at least $1 - e^{\Omega(\log^2 m)}$. On the other hand, we observe

$$\nabla_{\boldsymbol{w}} J_\upsilon(\boldsymbol{\theta}, \boldsymbol{w}; \widetilde{s}) = (\delta(s, a, s'; \boldsymbol{\theta}) - f(s, \boldsymbol{w})) \nabla_{\boldsymbol{w}} f(s, \boldsymbol{w}). \tag{102}$$

Following the same arguments as before yields $\|\nabla_{\boldsymbol{w}} J_\upsilon(\boldsymbol{\theta}, \boldsymbol{w}; \widetilde{s})\|_2 = \mathcal{O}(B \vee \log m)$ with high probability. The proof is concluded.

## D.2 Proof of Lemma D.2

Observe that

$$\|\nabla_{\boldsymbol{\theta}} J_\upsilon(\boldsymbol{z}; \widetilde{s}) - \nabla_{\boldsymbol{\theta}} J_\upsilon(\boldsymbol{z}'; \widetilde{s})\| \leq \|f(s, \boldsymbol{w})\nabla\delta(s, a, s'; \boldsymbol{\theta}) - f(s, \boldsymbol{w}')\nabla\delta(s, a, s'; \boldsymbol{\theta}')\|_2$$
$$+ \upsilon \|f(s, \boldsymbol{\theta})\nabla f(s, \boldsymbol{\theta}) - f(s, \boldsymbol{\theta}')\nabla f(s, \boldsymbol{\theta}')\|_2.$$

As we have

$$\|f(s, \boldsymbol{w})\nabla\delta(s, a, s'; \boldsymbol{\theta}) - f(s, \boldsymbol{w}')\nabla\delta(s, a, s'; \boldsymbol{\theta}')\|_2$$
$$\leq |f(s, \boldsymbol{w}) - f(s, \boldsymbol{w}')|\|\nabla\delta(s, a, s'; \boldsymbol{\theta})\|_2 + |f(s, \boldsymbol{w}')|\|\nabla\delta(s, a, s'; \boldsymbol{\theta}) - \nabla\delta(s, a, s'; \boldsymbol{\theta}')\|_2$$
$$\overset{(a)}{\leq} \mathcal{O}(1)\|\boldsymbol{w} - \boldsymbol{w}'\|_2 + \mathcal{O}(B^{4/3}m^{-1/6}(\log m)^{3/2}),$$

where (a) holds with probability at least $1 - e^{\Omega(\log^2 m)}$ by using [Allen-Zhu et al., 2019b, Theorem 1] on the first term and [Gao et al., 2019, Lemma A.5] on the second term. Similarly, we have

$$\|f(s, \boldsymbol{\theta})\nabla f(s, \boldsymbol{\theta}) - f(s, \boldsymbol{\theta}')\nabla f(s, \boldsymbol{\theta}')\|_2 \leq \mathcal{O}(1)\|\boldsymbol{\theta} - \boldsymbol{\theta}'\|_2 + \mathcal{O}(B^{4/3}m^{-1/6}(\log m)^{3/2}).$$

On the other hand, we have

$$\|\nabla_{\boldsymbol{w}} J_\upsilon(\boldsymbol{z}; \widetilde{s}) - \nabla_{\boldsymbol{w}} J_\upsilon(\boldsymbol{z}'; \widetilde{s})\|_2$$
$$= \|(\delta(s, a, s'; \boldsymbol{\theta}) - f(s, \boldsymbol{w}))\nabla f(s, \boldsymbol{w}) - (\delta(s, a, s'; \boldsymbol{\theta}') - f(s, \boldsymbol{w}'))\nabla f(s, \boldsymbol{w}')\|_2$$
$$\leq |\delta(s, a, s'; \boldsymbol{\theta}) - f(s, \boldsymbol{w}) - (\delta(s, a, s'; \boldsymbol{\theta}') - f(s, \boldsymbol{w}'))| \|\nabla f(s, \boldsymbol{w})\|_2 \tag{103}$$
$$+ |\delta(s, a, s'; \boldsymbol{\theta}') - f(s, \boldsymbol{w}')| \|\nabla f(s, \boldsymbol{w}) - \nabla f(s, \boldsymbol{w}')\|_2.$$

Using the same arguments as before, we can show

$$\|\nabla_{\boldsymbol{w}} J_\upsilon(\boldsymbol{z}; \widetilde{s}) - \nabla_{\boldsymbol{w}} J_\upsilon(\boldsymbol{z}'; \widetilde{s})\|_2 \leq \mathcal{O}(1)\|\boldsymbol{z} - \boldsymbol{z}'\|_2 + \mathcal{O}(B^{4/3}m^{-1/6}(\log m)^{3/2}).$$

This concludes our proof.

## D.3 Proof of Lemma D.3

Observe that

$$f(s, \boldsymbol{\theta}) - \widehat{f}(s, \boldsymbol{\theta}) = \int_0^1 \langle \nabla f(s, (1-t)\boldsymbol{\theta}_0 + t\boldsymbol{\theta}) - \nabla\widehat{f}(s, \boldsymbol{\theta}_0), \boldsymbol{\theta} - \boldsymbol{\theta}_0 \rangle \mathrm{d}t$$

$$\leq B \int_0^1 \|\nabla f(s, (1-t)\boldsymbol{\theta}_0 + t\boldsymbol{\theta}) - \nabla f(s, \boldsymbol{\theta}_0)\|_2 \mathrm{d}t = \mathcal{O}(B^{4/3}m^{-1/6}\sqrt{\log m}),$$

where the last equality is due to [Gao et al., 2019, Lemma A.5] and it holds with probability at least $1 - e^{\Omega(B^{2/3}m^{2/3})}$. This concludes the proof of the lemma.

### D.4 Proof of Lemma D.4

Following the proof from our previous Lemma A.1, we observe that

$$\|\widehat{e}_k^{(1)}\|_2^2 \leq 2\, \mathbb{E}_\mu[E_k^{(1,1)} + \upsilon^2 E_k^{(1,2)}],$$

with

$$E_k^{(1,1)} \leq 2|f(s, \boldsymbol{w}_k) - \widehat{f}(s, \boldsymbol{w}_k)|^2\, \|\nabla\bar{\delta}(s, \boldsymbol{\theta}_k)\|_2^2 + 2|\widehat{f}(s, \boldsymbol{w}_k)|^2\, \|\nabla\bar{\delta}(s, \boldsymbol{\theta}_k) - \nabla\widehat{\delta}(s, \boldsymbol{\theta}_k)\|_2^2,$$

$$E_k^{(1,2)} \leq 2|f(s, \boldsymbol{\theta}_k) - \widehat{f}(s, \boldsymbol{\theta}_k)|^2 \|\nabla f(s, \boldsymbol{\theta}_k)\|_2^2 + 2|\widehat{f}(s, \boldsymbol{\theta}_k)|^2 \|\nabla f(s, \boldsymbol{\theta}_k) - \nabla\widehat{f}(s, \boldsymbol{\theta}_k)\|_2^2.$$

Using Lemma D.3 and [Gao et al., 2019, Lemma A.5], with probability at least $1 - e^{\Omega(B^{2/3}m^{2/3})}$, we have

$$|f(s, \boldsymbol{\theta}_k) - \widehat{f}(s, \boldsymbol{\theta}_k)|^2\|\nabla f(s, \boldsymbol{\theta}_k)\|_2^2 = \mathcal{O}(B^{8/3}m^{-1/3}\log m).$$

Moreover, as $\nabla\widehat{f}(s, \boldsymbol{\theta}_k) = \nabla\widehat{f}(s, \boldsymbol{\theta}_0)$ and $\nabla\widehat{\delta}(s, \boldsymbol{\theta}_k) = \nabla\widehat{\delta}(s, \boldsymbol{\theta}_0)$, using (101), we have

$$|\widehat{f}(s, \boldsymbol{w}_k)|^2 \|\nabla\bar{\delta}(s, \boldsymbol{\theta}_k) - \nabla\widehat{\delta}(s, \boldsymbol{\theta}_k)\|_2^2 = \mathcal{O}\big((B \vee \log m)B^{2/3}m^{-1/3}\log m\big),$$

where the equality holds with probability at least $1 - e^{\Omega(\log^2 m)}$ due to [Allen-Zhu et al., 2019b, Theorem 1] and [Gao et al., 2019, Lemma A.5]. Combining the above yields

$$\|\boldsymbol{e}_k^{(1)}\|_2^2 = \mathcal{O}(B^{8/3}m^{-1/3}(\log m)^2). \tag{104}$$

On the other hand,

$$\|\boldsymbol{e}_k^{(2)}\|_2^2 \leq 2\mathbb{E}_\mu\left[|\bar{\delta}(s, \boldsymbol{\theta}_k) - f(s, \boldsymbol{w}_k) - (\widehat{\delta}(s, \boldsymbol{\theta}_k) - \widehat{f}(s, \boldsymbol{w}_k))|^2\|\nabla f(s, \boldsymbol{w}_k)\|_2^2\right]$$
$$+ 2\mathbb{E}_\mu\left[|\widehat{\delta}(s, \boldsymbol{\theta}_k) - \widehat{f}(s, \boldsymbol{w}_k)|^2\|\nabla\widehat{f}(s, \boldsymbol{w}_k) - \nabla f(s, \boldsymbol{w}_k)\|_2^2\right].$$

Likewise, we can bound

$$|\bar{\delta}(s, \boldsymbol{\theta}_k) - f(s, \boldsymbol{w}_k) - (\widehat{\delta}(s, \boldsymbol{\theta}_k) - \widehat{f}(s, \boldsymbol{w}_k))|^2\|\nabla f(s, \boldsymbol{w}_k)\|_2^2 = \mathcal{O}(B^{8/3}m^{-1/3}\log m)$$
$$|\widehat{\delta}(s, \boldsymbol{\theta}_k) - \widehat{f}(s, \boldsymbol{w}_k)|^2\|\nabla\widehat{f}(s, \boldsymbol{w}_k) - \nabla f(s, \boldsymbol{w}_k)\|_2^2 = \mathcal{O}\big((B \vee \log m)B^{2/3}m^{-1/3}\log m\big),$$

with probability at least $1 - e^{\Omega(\log^2 m)}$. Thus, we also get

$$\|\boldsymbol{e}_k^{(1)}\|_2^2 = \mathcal{O}(B^{8/3}m^{-1/3}(\log m)^2). \tag{105}$$

This concludes our proof.