[Reviews · NeurIPS 2020]

Review 1

Summary and Contributions: This work considers off-policy policy evaluation via a gradient temporal difference (GTD) algorithm when the function parameterization is a neural network (NN). The algorithm is developed in order to optimize the mean squared Bellman error (MSBE). By considering a saddle point reformulation, then a stochastic variant of primal-dual method is developed, where each is defined by an over-parameterized neural network. Global convergence of the proposed algorithm is established when the parameterization is a 2-layer ReLU NN architecture with m neurons as the number of neurons goes to infinity.

Strengths: The work considers a clearly important setting of off-policy evaluation, a canonical task in reinforcement learning that is used in imitation learning, and using prior experience to inform future policy improvement. Since neural parameterizations are commonly used in DRL today, the authors analyze how a gradient temporal difference scheme converges under various technical settings related to neural parameterization, and establish its convergence to global optimality. This is a substantial technical result that requires challenging convergence theory to be developed, which in my view is the main merit of the work.

Weaknesses: The philosophy of establishing convergence guarantees for neural networks under specific numbers of neurons is strange, because the number of neurons is a very coarse description of a network that can already be established by nonparametric estimators, i.e., Cho, Youngmin, and Lawrence K. Saul. "Kernel methods for deep learning." Advances in neural information processing systems. 2009. And numerous follow up works. Therefore, if the neural network analysis is to refine this approach, then it must also specify the *inter-layer* relationships and broader architectural choices to actually be useful to practitioners. As is, I don't see how the m of Lemma 4.1 can actually be used to inform choice of a neural architecture in any sharper manner than, e.g., a single layer RBF network. Also, reformulating Bellman's equations into saddle point problems has been previously studied: Shapiro, A. (2011). Analysis of stochastic dual dynamic programming method. European Journal of Operational Research, 209(1), 63-72. Dai, Bo, Niao He, Yunpeng Pan, Byron Boots, and Le Song. "Learning from conditional distributions via dual embeddings." In Artificial Intelligence and Statistics, pp. 1458-1467. 2017. B. Dai, A. Shaw, L. Li, L. Xiao, N. He, Z. Liu, J. Chen, and L. Song. Sbeed: Convergent reinforcement learning with nonlinear function approximation. In International Conference on Machine Learning, pages 1125–1134, 2018. The authors need to do a better job of explaining what is the technical similarity/difference between existing attempts to reformulate Bellman's equations into saddle point problems and that which is considered here. In my understanding, the only departing point is explicit theoretical analysis of how the algorithms for solving it interact with the neural parameterization, and establishing global convergence rather than local convergence. Given this aspect, I would be curious to see if there is any experimental corroboration of the phenomenon that enlarging the parameterization actually improves algorithm stability. I would expect then opposite to be true, because as the number of parameters increases, then the number of parameters one needs to estimate increases, and therefore algorithm convergence should *slow down.* These tradeoffs are not well-discussed in this work. For instance, I would be curious to see how H1 contrasts with a standard CNN or RNN architecture commonly used in DRL. Also, the importance and meaning of the algorithm being off-policy versus on-policy is not discussed. Are there any technical challenges or experimental merits to this setting? Why is it important? Convergence guarantees for this setting are discussed, but its inherent meaning is not.

Correctness: The claims and convergence theory seems well-justified and connected to existing recent trends in global optimality characterizations of algorithms for deep RL. But there is no experimentation so I decline to comment on empirical methodology.

Clarity: The paper is clear enough, although extremely densely packed so it is difficult to glean key importance of these details. Moreover, in appendices, there could be more connective discussion of how the different lemmas/propositions, etc. fit into the proofs, or a sketch of the logic pursued at the outset of the proofs.

Relation to Prior Work: Reference to finite-time performance of GTD is missing (section 5): Kumar, Harshat, Alec Koppel, and Alejandro Ribeiro. "On the sample complexity of actor-critic method for reinforcement learning with function approximation." arXiv preprint arXiv:1910.08412 (2019).

Reproducibility: No

Additional Feedback: I have read the rebuttal and the comments of the other reviewers. I would like to point out that the other reviewers may have interpreted aspects of the manuscript as novel that were not, especially related to global optimality results for neural parameterizations in policy gradient, and saddle point reformulations of Bellman's equations, as the authors have commented upon in the rebuttal. Therefore, the main technical novelty is that (i) they consider the off-policy setting (ii) they have established convergence that alleviates the rate dependence on the number of neurons by focusing on convergence in terms of L^2 metric. This is interesting, but is it enough to warrant acceptance? Is there any experimental importance/manifestation of alleviating this dependence? The paper has not honestly presented the merit of their contribution in this light, and is abstrusely written so it is very difficult to discern. I feel they are trying to bombard the reader into submission with mathematical details, rather than honestly contextualize the importance of their convergence results in theory and practice. (iii) Regarding the experimental importance, it is unacceptable that a paper about convergence results for neural parameterizations does not demonstrate the tradeoffs of architectural choices on algorithm performance, as this is the most difficult and critical aspect to making DRL work for new application domains. For these reasons, I will keep my opinion as marginally below the acceptance threshold.


Review 2

Summary and Contributions: The paper proposes a gradient temporal difference algorithm to minimize the mean squared Bellman error for policy evaluation. The authors analyze its convergence while using a two-layer neural network for function approximation and show that an approximately globally optimal solution is reached when the NN tends to become infinitely wide.

Strengths: The key strength of the work, and its difference from predecessors, is that the proposed algorithm does not involve computing the hessians of the NN which should make it scale better in practice.

Weaknesses: Please see my comment below.

Correctness: Yes, it seems to be the case although I have not gone over all the proofs in the appendix.

Clarity: Yes, the paper is written clearly and the authors have made a significant effort to structure it in a way which would make it easy to read and understand.

Relation to Prior Work: Yes

Reproducibility: Yes

Additional Feedback: I understand that the paper is primarily a theory paper but whenever an algorithm is proposed for a task, it is good to see some experiments demonstrating its effectiveness against baselines. It would be great if some experiments can be conducted to showcase the strengths of the algorithm better. --- Post-rebuttal comments --- After reading the author response, other reviews and having a discussion with other reviewers, I think there is definitely some merit to the points R1 brings up. However, though the paper's novelty might have been a bit more limited than it appeared to be at the first glance, the paper is still very interesting and technically the results are correct and contribute well to the literature. The authors have also made a reasonable effort to address the reviewers' concerns. I believe that the paper can still be accepted with changes to the final draft which: (a) address R1's points about the paper's contribution and novelty, and (b) include their new experiment from the rebuttal. So I am still keeping my original score of 7.


Review 3

Summary and Contributions: This is a theoretical paper which describes a GTD algorithm using two neural networks and off-policy samples which converges to a global optimum of the mean-squared Bellman error loss function given assumptions which are reasonable for a theoretical paper.

Strengths: - The paper is well presented, and the complex mathematical results are augmented with effective explanatory text. - The result is important and interesting, and the approach is creative. The derivation of the loss function as a mini-max problem was thought-provoking.

Weaknesses: - Some readers will dislike that it is a theoretical paper, and an algorithm is presented without it being tested or demonstrated. Without any application, it is not clear if this at all works or is relevant as assumptions are relaxed to a realistic setting or if the single-layer neural networks of the theorems are replaced with something more empirically successful.

Correctness: - I believe so. It is a dense theoretical derivation, and I cannot claim to have walked through or understood every step of the proofs. In parts, I leaned on the exposition, which made each step seem reasonable.

Clarity: Yes, the paper is quite well written and organized.

Relation to Prior Work: The paper nicely describes its influences in terms of neural network analysis and reinforcement learning. I would appreciate more exposition and less of a listing, but I'm not sure what I would cut to make room.

Reproducibility: Yes

Additional Feedback: AFTER AUTHOR COMMENTS AND REVIEWER DISCUSSION: Some of Reviewer 1's comments are well-taken, and improved citations are needed. I urge the authors to more clearly state some of this context. This lessens my enthusiasm somewhat, but I still feel this is an excellent paper.

[Author Response · NeurIPS 2020]

We would like to thank the three reviewers for their feedback. Upon acceptance, we will include (a) *a preliminary*
*experiment results on Neural GTD*, (b) *an expanded discussion on the theoretical results and relation to prior works*.
We first discuss the concern about experiments shared by **reviewer 1**, **reviewer 2**, **reviewer 3**.

●●● **Experiments**: The main motivation and contribution of this work have been theoreti-
cal – proving that off-policy learning using neural network (NN) functions approximation
can learn the value function with (almost) zero MSBE together with a finite time conver-
gence bound. However, we fully agree with the reviewers that a numerical experiment
would strengthen this claim. As a preliminary study, we consider an MDP taken from
the Garnet class with $|S| = 500$ states, $|A| = 5$ possible actions per state with uni-
formly distributed rewards, and the discount factor is $\gamma = 0.9$. We generate two random
policies with the same support as the behavior/target policies. We compare the average
MSBE against the number of neurons $m$ (for 2-layer, ReLU NN with random init.) after

$T = 2 \times 10^5$ iterations of neural GTD and neural TD [Cai et al., 2019] from 3 independent run of state/action. From the
figure, the average MSBE decreases with $m$ stably for neural GTD, while it fluctuates with $m$ with neural TD, indicating
that the latter can be unstable in the off-policy setting ([Cai et al., 2019] analyzed the neural TD for on-policy). We will
include simulations that averages with more trajectories to obtain the expected performance.

**Reviewer 1**: We thank you for the review and constructive comments. See the point-to-point response below.

**NN Architecture**: As discussed in (4) and the abstract, our analysis are based on a **2-layer, fully connected, ReLU**
NN with $d$-dimensional input, and $m$ hidden neurons. Having said that, it is an interesting future direction to analyze
neural GTD with other types of NN architecture. Lastly, our analysis is based on the surrogate (linearized) NN function
(14)-(16), which is akin to a kernel approximation (called the Neural Tangent Kernel, see [Jacot et al., 2018]). In the
final version, we will discuss about these connections in detail.

**Saddle-point Reformulation**: Just as the reviewer said, the reformulation of the min MSBE problem as saddle point
optimization follows from prior work such as [Dai et al., 2017,2018], [Shapiro 2011]. We do not claim this as our
main contribution either. Instead, the introduction of the **dual NN** in (9) is new. Particularly, a simple application of
saddle point reformulation to min MSBE results in (8), which involves an $|S|$-dim. sub-problem. As $|S|$ is large (can be
infinite), the dual NN is used to circumvent this intractability. We will include your references in the final version.

Dai et al. [2018] derived a similar reformulation to our paper. Besides only guaranteeing convergence to a stationary
point (whilst we showed convergence to a global MSBE minimizer for neural GTD), their approximation error
is characterized by the $\ell_\infty$-norm (Theorem 7). The $\ell_\infty$-norm requirement is restrictive as it requires the function
approximation to be **uniformly accurate**, i.e., $\min_\theta |V^\pi(s) - \hat{V}(s;\theta)|$ is small for **every** $s \in S$. On the other hand, we
only require a small $L^2$ variation (H4), i.e., $\min_\theta \mathbb{E}_s[|V^\pi(s) - \hat{V}(s;\theta)|^2]$. Lastly, their algorithm requires a computation
oracle which finds an **exact** solution to the inner optimization (see line 7 of their algorithm 1), which can be intractable.
E.g., if an NN approximation is used, this oracle will need to solve an NN regression problem.

**Technical Contributions**: On top of providing an explicit theoretical analysis with convergence to a *global minimum*
MSBE, we emphasize on the technical novelty developed in our paper – We show the convergence rates of the $L^2$
variation w.r.t. the optimal NN over the **function space** (see (18)) that is *independent of the no. of neurons $m$*. This
is different from existing analysis on GTD learning with linear function approximation (LFA), which show the mean
square error of parameter in Euclidean norm. Moreover, the latter analysis may not work if the LFA used involve
$p \to \infty$ parameters, since in this case the expected GTD update matrix may cease to be Hurwitz. On the other hand, the
convergence rate measured in this $L^2$ variation (as well as MSBE) is unaffected by the no. of parameters; as seen in
Theorem 3.1-3.4 and the numerical experiments above.

For a comparison, as analyzed in [Dalal et al. 2019], GTD with LFA finds an optimal parameter at $\mathcal{O}(1/k)$ (w.h.p.),
while neural GTD's rate is $\mathcal{O}(\log k/\sqrt{k})$. However, as mentioned the rate of GTD with LFA is valid only for finite $p$.
Besides, the reference [Kumar et al. 2019] is on actor-critic for policy improvement. Though the subroutine of GTD
with LFA is used as the critic, the analysis therein is not directly comparable to this work.

**Off-policy Learning**: This is when the states/actions received during policy evaluation follow a *behavior policy* which
is different from the *target policy* that we want to evaluate. It is a common setting, e.g., when only one set of data
is available and our goal is to evaluate different policies without gathering more data. Our study is important as it is
known that classical TD learning with LFA can diverge in off-policy learning [Sutton et al. 2009b], and by extension
the neural TD may diverge as well. Particularly we show neural GTD is still efficient for off-policy.

**Reviewer 2**: We thank you for the positive comments. As mentioned, we will now provide a small numerical experiment
to strengthen our claims in the final version.

**Reviewer 3**: We thank you for the positive comments. In the final version, we will extend the discussions of the
theoretical results and relation to prior works. We also provide a small numerical experiment to verify our claims (see
above). In addition, we are considering to extend the analysis for more general NN architectures.

[Meta-Review · NeurIPS 2020]

This paper generated substantial discussion from the reviewers. Reviewer 1's points of lack of contextualization are well-taken by the other reviewers. That said, the meta-reviewer (in consultation with the Senior Area Chair) agrees that the theoretical contribution will be of interest to the NeurIPS community, and the clarity & sharpness of the authors' response suggests the authors are quite capable of revising the paper to more clearly discuss context and articulate their contribution. As such, the metareviewer is recommending accept.